# Adapting to Linear Separable Subsets with Large-Margin in Differentially Private Learning

**Erchi Wang**[1]   **Yuqing Zhu**[2]   **Yu-Xiang Wang**[1]

## Abstract

This paper studies the problem of differentially private empirical risk minimization (DP-ERM) for binary linear classification. We obtain an efficient $(\varepsilon, \delta)$-DP algorithm with an empirical zero-one risk bound of $\tilde{O}\left(\frac{1}{\gamma^2 \varepsilon n} + \frac{|S_{\text{out}}|}{\gamma n}\right)$ where $n$ is the number of data points, $S_{\text{out}}$ is an arbitrary subset of data one can remove and $\gamma$ is the margin of linear separation of the remaining data points (after $S_{\text{out}}$ is removed). Here, $\tilde{O}(\cdot)$ hides only logarithmic terms. In the agnostic case, we improve the existing results when the number of outliers is small. Our algorithm is highly adaptive because it does not require knowing the margin parameter $\gamma$ or outlier subset $S_{\text{out}}$. We also derive a utility bound for the advanced private hyperparameter tuning algorithm.

## 1. Introduction

We investigate differentially private empirical risk minimization (DP-ERM) (Chaudhuri et al., 2011; Bassily et al., 2014) under the setting of learning large-margin halfspaces. In classification problems, algorithms that create decision boundaries with larger separations between classes, a.k.a. large margins, tend to have stronger generalization performance (Vapnik, 1998; Panagiotakopoulos & Tsampouka, 2011). This principle has been leveraged in many classical machine learning algorithms, such as the Perceptron (Rosenblatt, 1958; Novikoff, 1962), AdaBoost (Freund & Schapire, 1997), and Support Vector Machine (Vapnik, 1998; Cortes & Vapnik, 1995). It is natural to ask whether we can design a *data-adaptive DP algorithm* to benefit from a large-margin condition. This is challenging because differential privacy requires the algorithm's output to be "close enough" when two datasets differ by one data point (Dwork et al., 2006;

McSherry & Talwar, 2007a; Dwork et al., 2014). A single point change can cause the margin to shift drastically, from a positive value to zero or the reverse, as demonstrated in Figure 1. Despite the theoretical challenge, empirically, a larger margin is believed to be the reason why pre-trained features help private learners to work better (De et al., 2022).

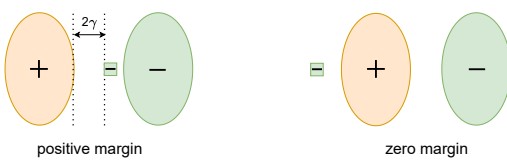

*Figure 1.* Margin is unstable after changing one point (marked by the green square), thus hard to design a data-dependent DP-mechanism using standard techniques (e.g., smoothed sensitivity (Nissim et al., 2007) or propose-test-release (Dwork & Lei, 2009)).

To validate this hypothesis, we evaluated the margin of SVM classifiers trained on the CIFAR-10 dataset using pre-trained features from Vision Transformer (ViT) (Dosovitskiy et al., 2021) and ResNet-50 (He et al., 2016). Interestingly, as reported in Figure 2, while the margin remains at zero for the whole dataset, if we allow removing a few "trouble makers" (misclassified or other points near the decision boundary), it becomes clear that ViT-based features achieve a larger margin and increase more quickly than ResNet-50-based features as removing more outliers.

After approximately $0.1\%$ of points are removed, the remaining data become linearly separable for both ViT and ResNet-50-based features. This phenomenon also relates to neural collapse theory (Papyan et al., 2020; Wang et al., 2024), which states that the last layer of a deep neural network trained on $K$-class classification task converges to $K$ distinct points. The geometric margin (Eq. 1) does not capture this phenomenon. Since the CIFAR-10 dataset is not linearly separable, the geometric margin is zero. This inspires us to ask:

*Can we design an efficient differentially private algorithm for agnostic learning halfspaces while adapting to linear separable subsets with large margins?*

---

[1]Halıcıoğlu Data Science Institute, UC San Diego [2]LinkedIn. Correspondence to: Yu-Xiang Wang <yuxiangw@ucsd.edu>, Erchi Wang <erw011@ucsd.edu>.

*Proceedings of the 42$^{nd}$ International Conference on Machine Learning*, Vancouver, Canada. PMLR 267, 2025. Copyright 2025 by the author(s).

*Table 1.* Summary of the results on the population zero-one loss (with high probability). *Realizable case* means that the data is linearly separable with *geometric margin* at least $\gamma$. In the *agnostic case*, $\gamma$ is the data-independent *confidence margin* parameter in Row 2 and 3. In Row 4, $\gamma$ denotes the geometric margin of a data subset after $S_{\text{out}}$ is removed. For a clean comparison, we assume $\|\mathbf{x}\|, \|\mathbf{w}\|$ to be $\mathcal{O}(1)$, the privacy loss $\varepsilon \leq 1$, and omit the dependence on factors of order $\text{polylog}(n, \varepsilon, 1/\delta, 1/\beta)$; $\tilde{L}_S^\gamma(\cdot)$ ($\tilde{\mathcal{R}}_S^\gamma(\cdot)$) denotes average of empirical $\gamma$-hinge loss (empirical $\gamma$-ramp loss).

| Source | Realizable case | Agnostic case | polynomial-time? |
|---|---|---|---|
| Nguyễn et al. (2020, Thm. 6, Thm. 11) | $\dfrac{1}{n\gamma^2\varepsilon}$   known $\gamma$ | NA | ✓ |
| Bassily et al. (2022, Thm. 3.1) | $\dfrac{1}{n\gamma^2\varepsilon}$ | $\dfrac{1}{n\gamma^2\varepsilon} + \min\limits_{w \in \mathcal{B}^d(1)} \left( \tilde{\mathcal{R}}_S^\gamma(w) + \sqrt{\left(\dfrac{1}{n^2\gamma^2} + \dfrac{1}{n}\right) \cdot \tilde{\mathcal{R}}_S^\gamma(w)} \right)$ | ✗ |
| Bassily et al. (2022, Thm. 3.2) | $\dfrac{1}{n^{1/2}\gamma\varepsilon^{1/2}}$ | $\dfrac{1}{n^{1/2}\gamma\varepsilon^{1/2}} + \min\limits_{w \in \mathcal{B}^d(1)} \tilde{L}_S^\gamma(w)$ | ✓ |
| Theorem 4.1 | $\dfrac{1}{n\gamma^2\varepsilon}$ | $\min\limits_{\substack{S_{\text{out}} \subset S \\ \gamma := \text{margin}(S \backslash S_{\text{out}})}} \left( \dfrac{|S_{\text{out}}|}{n\gamma} + \dfrac{1}{n\gamma^2\varepsilon} \right)$ | ✓ |

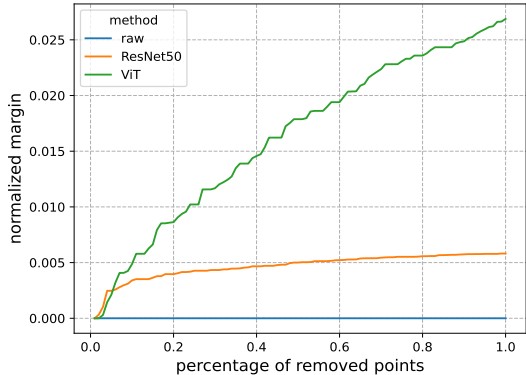

*Figure 2.* The number of removed points (in percentage of $n$) vs normalized margin. Classes 1 and 9 from the CIFAR10 training set are used. As more points are removed, the margin increases. We include more implementation details in Appendix N.

We give an affirmative answer to this question and summarize our contribution here.

## 1.1. Our Contribution

**An efficient algorithm adapting to large margins** We propose an efficient algorithm adapting to large margin subsets differential privately, without the need for any tuning parameters nor assumptions of realizability, as stated in Algo. 3.

**Inlier-outlier analysis of gradient descent** We propose margin inlier/outlier (Definition 6.1), as a generalization of geometric margin. We obtain data adaptive upper bounds on both empirical and population risk, which depend on the number of margin outliers (Theorem 4.1). This analysis

technique allows us to achieve a $\sqrt{n}$ improvement in the population risk bound compared to Theorem 3.2 in (Bassily et al., 2022). Moreover, the term $|S_{\text{out}}|/\gamma n$ naturally adapts to the problem's complexity, allowing our results to extend to the agnostic case, where directly applying Theorem 6 from (Nguyễn et al., 2020) is not feasible. Finally, this analysis technique enables us to extend to unbounded parameter space. This avoids the assumption on the bounded domain as stated in (Nguyễn et al., 2020; Bassily et al., 2022).

## 1.2. Related Work

**Prior work on DP large-margin learning.** To the best of our knowledge, Nguyễn et al. (2020) is the first work to achieve a dimension-free excess population risk for differentially private learning of large-margin halfspaces. Under the assumptions of linear separability and known geometric margin, they are able to achieve a fast rate of $\tilde{\mathcal{O}}\left(\frac{1}{n\gamma^2\varepsilon}\right)$ using an efficient approximate DP algorithm. However, in practice, the linear separability of the data and the geometric margin value are often unknown without directly accessing the data. To address this, Bassily et al. (2022) uses the confidence margin (Mohri et al., 2018), a data-independent quantity, and achieves an $\tilde{O}\left(\frac{1}{\gamma\sqrt{n}} + \frac{1}{\gamma^2 n\varepsilon}\right)$ excess empirical risk on surrogate loss in the agnostic case.

Pure-DP algorithms are also discussed in Nguyễn et al. (2020) and Bassily et al. (2022). However, they suffer from computational inefficiency due to the use of exponential mechanisms on candidate sets of size $\mathcal{O}(\exp(1/\gamma))$. For completeness, we include these results for both the agnostic and realizable cases and present a comparison with our result in Table 1.

Bun et al. (2020) and Ghazi et al. (2021) study private learning of large-margin halfspaces with an emphasis on

robustness. Both two works make specific assumptions about realizability and noise oracles. Moreover, proposed algorithms require taking the geometric margin as input, i.e., Bun et al. (2020, Algo. 4) and Ghazi et al. (2021, Algo. 1).

For earlier works with $\text{poly}(d)$ dependence in population risk, we refer the readers to Section 1.2 of Nguyễn et al. (2020) for a comprehensive survey.

**Private hyperparameter tuning.** To achieve the desired results, the margin parameter also needs to be selected. Bassily et al. (2022) addresses this by privately selecting the optimal confidence margin using the generalized exponential mechanism (Raskhodnikova & Smith, 2016). However, their approach relies on a score function that incorporates the minimized empirical risk over the hypothesis space, which is generally hard to compute (Bassily et al., 2022, Lemma F.3).

In addition to classical differentially private selection mechanisms such as the exponential mechanism (McSherry & Talwar, 2007b), the report-noisy-max (Dwork et al., 2014), and the sparse vector technique (Dwork et al., 2009; Zhu & Wang, 2020), there have been notable advancements in private hyperparameter tuning. This research direction began with (Liu & Talwar, 2019) and was further refined by (Papernot & Steinke, 2022), which utilized Rényi differential privacy, and by (Koskela & Kulkarni, 2024), which incorporated subsampling techniques. More recently, (Koskela et al., 2024) further improves the practicality by providing tighter privacy accounting through the use of privacy profiles. These advanced methods allow the final privacy budget to grow logarithmically with the number of repetitions, as opposed to the (sub)linear growth observed with (advanced) naive composition (Dwork et al., 2014).

We examine two hyperparameter tuning methods: a simple brute-force approach (Algo. 2) and an advanced private tuning method (Algo. 4). Their utility and computational efficiency are compared in Section 7.

## 2. Preliminary

**Symbols and notations.** We use boldface letters (e.g., $\mathbf{x}$) to denote vectors. Calligraphic letters are used as follows: $\mathcal{F}$ represents the function class, $\mathcal{H}$ denotes the classifier class, $\mathcal{X}$ represents the data universe, and $\mathcal{A}$ denotes a random algorithm. The notation $\mathbb{E}_{\mathcal{A}}[\cdot]$ indicates the expectation taken over the randomness of algorithms. The binary operation $\cdot \wedge \cdot$ denotes taking the minimum between the two inputs. Throughout this paper, $\langle \cdot, \cdot \rangle$ represents the inner product, and $\| \cdot \|$ denotes the induced $L_2$ norm. The set $\mathcal{B}^d(r) := \{\mathbf{x} \in \mathbb{R}^d \mid \|\mathbf{x}\| \leq r\}$ is defined as the $L_2$ ball in $\mathbb{R}^d$ with radius $r$. The sign function is denoted by $\text{sign}(\cdot)$, and the indicator function is represented by $\mathbf{1}(\cdot)$. For a set $A \subset \mathbb{R}^d$, we use $|A|$ to denote its cardi-

nality. The $L_2$ projection operator onto $A$ is denoted as $\text{Proj}_A(\mathbf{x}) := \arg\min_{\mathbf{v} \in A} \|\mathbf{v} - \mathbf{x}\|$. The power set of $A$ is denoted by $2^A$. The probability generating function of a random variable $K$ is represented by $\text{PGF}_K(\cdot)$. Throughout this paper, the notation $\mathcal{O}$ is used to suppress universal constants, while $\tilde{\mathcal{O}}$ hides $\text{polylog}(n, 1/\beta, 1/\delta)$ factors.

**Differential Privacy.** We call two datasets $X, X' \in \mathcal{X}^n$ neighboring datasets if they differ up to one element.

**Definition 2.1** (Differential privacy (Dwork et al., 2006))**.** A randomized algorithm $\mathcal{M} : \mathcal{X}^n \to \Omega$ is $(\varepsilon, \delta)$-DP (differential private) for $\varepsilon, \delta \in (0, 1)$ if for any neighbouring datasets $X, X' \in \mathcal{X}^n$ and any measurable subset $O \subseteq \Omega$, $\mathcal{M}$ satisfies:

$$\mathbb{P}(\mathcal{M}(X) \in O) \leq e^\varepsilon \mathbb{P}(\mathcal{M}(X') \in O) + \delta$$

**Definition 2.2** (Gaussian Differential Privacy (GDP) (Dong et al., 2022))**.** We say a mechanism $\mathcal{M}$ satisfies $\mu$-GDP if for any neighbouring dataset $X, X' \in \mathcal{X}^n$,

$$H_{e^\varepsilon}(\mathcal{M}(X)\|\mathcal{M}(X')) \leq H_{e^\varepsilon}(\mathcal{N}(0,1)\|\mathcal{N}(\mu,1)), \ \forall \varepsilon \in \mathbb{R},$$

where $\mathcal{N}(0, 1)$ stands for standard normal distribution and $H_{e^\varepsilon}(\cdot\|\cdot)$ denotes the hockey-stick divergence between two probability distributions (Sason & Verdu, 2016).

*Remark* 2.3. While $(\varepsilon, \delta)$-DP offers a direct description of privacy leakage, it's messy and loose when composing multiple mechanisms together (Near & Abuah, 2021, Chapter 6). Instead, GDP not only has a clean form for composition: composing $\{\mathcal{M}_i\}_{i=1}^k$ mechanisms with each mechanism being $\mu_k$-GDP yields $\left(\sqrt{\sum_{i=1}^k \mu_i^2}\right)$-GDP, but offers a tight characterization for Gaussian mechanism: Gaussian mechanism with L2 sensitivity $\Delta$ and noise parameter $\sigma$ satisfies $\Delta/\sigma$-GDP. Thus, throughout this paper, we use GDP and its composition when possible, though the final result is still stated in $(\varepsilon, \delta)$-DP.

**Geometric margin.** We first define the margin of dataset $S$ with respect to a linear classifier $h_{\mathbf{w}} : (\mathbf{x}, y) \mapsto \text{sign}(y\langle \mathbf{w}, \mathbf{x} \rangle)$ to be:

$$\text{Margin}(\mathbf{w}; S) = \max \left\{ \min_{(\mathbf{x}, y) \in S} \frac{y\langle \mathbf{w}, \mathbf{x} \rangle}{\|\mathbf{w}\|}, 0 \right\}$$

We say dataset $S$ is linear separable if there exists some linear classifier $\mathbf{w}$ with $\gamma(\mathbf{w}; S) > 0$. The *geometric margin* for dataset $S$ is defined to be:

$$\gamma(S) := \max_{\mathbf{w} \in \mathbb{R}^d} \text{Margin}(\mathbf{w}; \mathcal{S}) \tag{1}$$

As shown in Figure 2, a dataset may initially have a zero margin. However, after removing certain "outliers," the remaining dataset can exhibit a positive margin. We formalize this concept and introduce the definitions of margin inliers and outliers in Definition 6.1.

## 3. Settings

**Loss function.** We use $\ell_c$ to represent hinge loss with confidence margin parameter $c$:

$$\ell_c(\mathbf{w}; (\mathbf{x}, y)) = \max\{0, 1 - y\langle \mathbf{w}, \mathbf{x}\rangle/c\}$$

$\tilde{L}_c(\mathbf{w}; S)$ $\left(\hat{L}_c(\mathbf{w}; S)\right)$ represents the averaged (summed) empirical hinge loss for estimator $\mathbf{w}$ over the dataset $S$.

*Remark* 3.1. Throughout this paper, the confidence margin is only used as a parameter for hinge loss. It is unrelated to the data-dependent quantities we defined, such as margin inliers/outliers (Definition 6.1).

In our optimization algorithm, we use hinge loss. It is important to note that varying confidence margin parameters can result in different loss values, even with the same estimator $\mathbf{w}$ and dataset $S$. To ensure a fair comparison, we express utility upper bounds using zero-one loss:

$$\ell_{0-1}(\mathbf{w}; (\mathbf{x}, y)) = \mathbf{1}\{y\langle \mathbf{w}, \mathbf{x}\rangle < 0\}$$

For an estimator $\mathbf{w}$, we denote its averaged empirical zero-one loss on dataset $S$ as $\tilde{\mathcal{R}}_S(\mathbf{w})$ and its summed version as $\hat{\mathcal{R}}_S(\mathbf{w})$. Furthermore, $\mathcal{R}_D(\mathbf{w})$ represents the population-level zero-one risk for the data distribution $D$.

**Assumptions.** We make the following assumptions: the training dataset $S = \{(\mathbf{x}_1, y_1), \ldots, (\mathbf{x}_n, y_n)\}$ is sampled i.i.d. from an unknown distribution $D$ defined on $\mathcal{B}^d(1) \times \{\pm 1\}$. We consider the function class $\mathcal{F} = \{\langle \cdot, \mathbf{w}\rangle \mid \mathbf{w} \in \mathbb{R}^d\}$ and the hypothesis class of classifiers $\mathcal{H} = \{\mathbf{x} \mapsto \text{sign}(f(\mathbf{x})) \mid f \in \mathcal{F}\}$. The assumption that $\mathbf{x}$ and $\mathbf{w}$ lie in the unit ball is made without loss of generality. In Section 8.1, we extend our results to the more general case where $\mathbf{x} \in \mathcal{B}^d(b)$ and $\mathbf{w} \in \mathbb{R}^d$.

**Problem Setup.** We address the problem of agnostic proper learning of halfspaces with differential privacy. Given a dataset $S$ and a privacy budget $\varepsilon, \delta$, our goal is to design an algorithm that takes $S$, $\varepsilon$, and $\delta$ as input and outputs a classifier $\mathbf{w}_{\text{out}}$ from the hypothesis class $\mathcal{H}$, ensuring a small (empirical) population zero-one risk ($\tilde{\mathcal{R}}_S(\mathbf{w}_{\text{out}})$) $\mathcal{R}_D(\mathbf{w}_{\text{out}})$.

## 4. Main Result

We state our adaptive margin bound as follows:

**Theorem 4.1.** *There exists an* efficient *algorithm $\mathcal{M}^*$ (Algo. 3), for any input dataset $S$, privacy budgets $\varepsilon, \delta$ satisfying $\delta \in (0, 1)$ and $\varepsilon \in (0, 8\log(1/\delta))$ such that: (1) $\mathcal{M}^*$ is $(\varepsilon, \delta)$-DP. (2) With high probability, for any $S_{\text{out}} \in 2^S$ with $\gamma := \gamma(S \setminus S_{\text{out}}) > 0$, simultaneously:*

$$\tilde{\mathcal{R}}_S(\mathcal{M}^*(S, \varepsilon, \delta)) \leq \tilde{\mathcal{O}}\left(\frac{1}{n\gamma^2 \min\{\varepsilon, 1\}} + \frac{|S_{\text{out}}|}{\gamma n}\right) \wedge 1$$

$$\mathcal{R}_D(\mathcal{M}^*(S, \varepsilon, \delta)) \leq \tilde{\mathcal{O}}\left(\frac{1}{n\gamma^2 \min\{\varepsilon, 1\}} + \frac{|S_{\text{out}}|}{\gamma n}\right) \wedge 1$$

The proof builds on Theorem 6.5 and Theorem 6.7, with the details deferred to Appendix H and Appendix I.

Let's make a few observations. In the realizable case, $\mathcal{M}^*$ naturally adapts to the margin without requiring prior knowledge of its value, in contrast to the approach proposed by Nguyễn et al. (2020, Theorem 6&11). In the agnostic setting, $S_{\text{out}}$ can be interpreted as outliers. When $|S_{\text{out}}|$ is sufficiently small $\left(\approx o(\sqrt{n})\right)$, our results achieve a $\sqrt{n}$ improvement over Bassily et al. (2022, Theorem 3.2). For a detailed comparison, see Table 1.

The algorithm $\mathcal{M}^*$ in Theorem 4.1 does not need any tuning parameters. It is highly adaptive as it can compete with the best $S_{\text{out}}^* \in 2^S$ that minimizes the bound without having to explicitly search for $S_{\text{out}}^*$. It also does not incur the typical statistical costs of $\log(2^n)$ of adaptivity to $2^S$, which would render the bound vacuous. In addition, $\mathcal{M}^*$ is also computationally efficient. We will see in the next section that $\mathcal{M}^*$ has a modular design, and it only requires invoking an off-the-shelf solver for the convex DP-ERM problem a few times.

## 5. Algorithms

In this section, we provide details for our algorithms. The algorithm $\mathcal{M}^*$ (Algo. 3) is designed as a composition of two components: the private hyperparameter tuning algorithm $\mathcal{A}_{\text{Iter}}$ (Algo. 2) and the noisy gradient descent algorithm with Johnson-Lindenstrauss projection $\mathcal{A}_{\text{JLGD}}$ (Algo. 1).

### 5.1. Base Algorithm: JL-Noisy Gradient Descent

To obtain a dimension-free rate, we apply the Johnson-Lindenstrauss (JL) projection to reduce the dimensionality while preserving the linear separability of projected data. The "adequate" projection dimension for maintaining separability is determined by the data margin. Informally:

**Lemma 5.1** (Informal version of Lemma B.2)**.** *Let dataset $S = \{(\mathbf{x}_i, y_i)\}_{i=1}^n \subset \mathbb{R}^d \times \{\pm 1\}$ satisfying $\gamma(S) > 0$, choosing $k = \mathcal{O}(1/\gamma^2(S))$ in the JL projection ensures that the projected dataset $\Phi S := \{(\Phi\mathbf{x}_i, y_i)\}_{i=1}^n$ satisfies $\gamma(\Phi S) > \frac{\gamma(S)}{2}$ with high probability.*

In Algo. 1, the data is first projected into a lower-dimensional space using a JL matrix $\Phi \in \mathbb{R}^{k \times d}$. Next, we further clip the $L_2$ norm of data using projection to bound the sensitivity. After running noisy gradient descent for $T$ iterations, we can choose to output either the last-iterate estimator $\mathbf{w}_{\text{out}}$ or the averaged estimator $\bar{\mathbf{w}}_{\text{out}} = \frac{1}{T-1}\sum_{t=0}^{T-1} \mathbf{w}_t$ (described in $\mathcal{A}_{\text{NGD}}$ (Algo. 5)). These two choices correspond to utility guarantees in expectation form or high probability form, as stated in the Theorem 6.5.

We note that using JL projection to preprocess data in pri-

vate learning of halfspaces has been explored in previous work such as Nguyễn et al. (2020); Bassily et al. (2022). Compared to Nguyễn et al. (2020, Algo. 1), we do not assume that the projection dimension $k$ is calculated from the actual data margin $\gamma(S)$, which we provide more details in Algo. 3. Compared to Bassily et al. (2022, Algo. 2), our projection dimension is chosen to be $\tilde{\mathcal{O}}(1/\gamma^2)$ instead of $\tilde{\mathcal{O}}(n\varepsilon)$. In addition, we use a full-batch gradient descent algorithm rather than SCO-type methods, which avoids the $1/\sqrt{n}$ term in the convergence rate.

---

**Algorithm 1:** $\mathcal{A}_{\mathsf{JLGD}}(\Phi, c, S, \mu)$

1 **Input:** Dataset $S = \{\mathbf{x}_i, y_i\}_{i=1}^n$ with $\|\mathbf{x}\| \leq v$,
  JL projection matrix $\Phi \in \mathbb{R}^{d \times k}$,
  Hinge loss parameter $c$, GDP budget $\mu$
2 $\Phi S := \{(\mathrm{Proj}_{\mathcal{B}^k(2v)}(\Phi \mathbf{x}_i), y_i)\}_{i=1}^n$
3 $\tilde{\mathbf{w}} = \mathcal{A}_{\mathsf{NGD}}(\ell_c, \Phi S, \mu)$ ▷ *Algo. 5*
4 **Output:** $\Phi^\top \tilde{\mathbf{w}}$

---

### 5.2. Construction of Main Algorithm $\mathcal{M}^*$

Our main algorithm $\mathcal{M}^*$ (Algo. 3) contains three steps:

**Step 1. (Line 2 of Algo. 3)** We begin by discretizing the range of possible margin values to create a set of margin parameters, denoted as $\Gamma$. This set is constructed as a logarithmic grid over the interval $[0, 1]$. The analysis of the approximation error introduced by this discretization is provided in Theorem 6.4.

**Step 2. (Line 4-7 of Algo. 3)** For each margin parameter $\gamma$ in the set $\Gamma$, we construct corresponding projection matrices $\Phi_\gamma$. The projection dimension $k_\gamma$ is set to $\tilde{\mathcal{O}}(1/\gamma^2)$ to ensure margin preservation, as demonstrated in Lemma 6.2. Importantly, the construction of each $\Phi_\gamma$ is independent of the data, meaning it does not consume any additional privacy budget.

---

**Algorithm 2:** $\mathcal{A}_{\mathsf{Iter}}(\mathcal{M}, \Theta, S, \mu)$

1 **Input:** Base mechanism $\mathcal{M}_{\mathrm{base}}$,
  hyperparameter set $\Theta$, dataset $S$,
  GDP budget $\mu$
2 **Choose:** Criterion function $U$ with sensitivity $\Delta$
3 **for** $\theta \in \Theta$ **do**
4 $\quad m_\theta = \mathcal{M}_{\mathrm{base}}(\theta; S, \frac{\mu}{\sqrt{2|\Theta|}})$
5 $\quad \xi \sim \mathcal{N}(0, \frac{2|\Theta|\Delta^2}{\mu^2})$
6 $\quad u_\theta = U(m_\theta; S) + \xi$
7 $\theta_{\mathrm{out}} = \arg\min_{\theta \in \Theta} u_\theta$
8 **Output:** $(m_{\theta_{\mathrm{out}}}, \theta_{\mathrm{out}})$

---

**Step 3. (Line 8 of Algo. 3)** We evaluate all margin configura-

tions in $\Gamma$ and identify the margin parameter that minimizes the zero-one risks. Given that the size of the margin grid is $\lceil \log_2(n) \rceil + 1$, we utilize a brute force approach that iterates through all configurations, as described in Algo. 2. To minimize empirical risk, we use the average empirical zero-one loss as the scoring function. We employ a penalized scoring function to minimize population risk, defined in Eq. 3.

The input to Algo. 3 includes only the dataset $S$ and the privacy parameters $\varepsilon$ and $\delta$. Since the margin grid $\Gamma$ is constructed in a data-independent manner, no prior knowledge of the data margin is required.

---

**Algorithm 3:** DP Adaptive Margin $\mathcal{M}^*(S, \varepsilon, \delta)$

1 **Input:** dataset $S = \{\mathbf{x}_i, y_i\}_{i=1}^n$, Privacy budget $\varepsilon, \delta$
2 **Set:** Margin grid $\Gamma = \{\frac{1}{n}, \frac{2}{n}, \frac{4}{n}, ..., \frac{2^{\lfloor \log_2 n \rfloor}}{n}, 1\}$,
  GDP budget $\mu = \frac{\varepsilon}{2\sqrt{2\log(1/\delta)}}$,
  failure probability for JL projection $\beta$
3 Initialize hyperparameter set $\Theta = \{\phi\}$ ▷ *empty set*
4 **for** $\gamma \in \Gamma$ **do**
5 $\quad k_\gamma = \mathcal{O}\left(\frac{1}{\gamma^2} \log(\frac{|\Gamma|(n+2)(n+1)}{\beta})\right)$
6 $\quad \Phi_\gamma \sim (\mathrm{Rad}(\frac{1}{2})/\sqrt{k_\gamma})^{k_\gamma \times d}$
7 $\quad \Theta = \Theta \cup \{(\gamma, \Phi_\gamma)\}$
8 $(\tilde{\mathbf{w}}_{\mathrm{out}}, \gamma_{\mathrm{out}}, \Phi_{\gamma_{\mathrm{out}}}) = \mathcal{A}_{\mathsf{Iter}}(\mathcal{A}_{\mathsf{JLGD}}(\cdot), \Theta, S, \mu)$
9 **Output:** $(\tilde{\mathbf{w}}_{\mathrm{out}}, \gamma_{\mathrm{out}}, \Phi_{\gamma_{\mathrm{out}}})$

---

For computational efficiency, Algo. 3 is constructed by running Algo. 1 for $\lceil \log_2(n) \rceil + 1$ times, making it a polynomial-time algorithm because Algo. 1 itself runs in polynomial time. Additionally, as shown in Section 6.1, our convergence analysis (Eq. 2) is compatible with any black-box optimization method, allowing the replacement of full-batch gradient descent with more efficient DP-ERM methods, such as DP-SGD, to improve computational efficiency further.

## 6. Proof Sketch

We provide privacy and utility analysis of our algorithms. To begin with, we define the margin inliers ($\mathcal{S}_{\mathrm{in}}$) and the margin outliers ($\mathcal{S}_{\mathrm{out}}$), which help extend the definition of the geometric margin.

The motivation for defining $\mathcal{S}_{\mathrm{out}}(\gamma)$ arises from the observation that after removing a few "outliers" denoted as $S_{\mathrm{out}}$, the remaining dataset $\gamma(S \setminus S_{\mathrm{out}})$ becomes $\gamma$-separable. To avoid the combinatorial explosion associated with selecting $S_{\mathrm{out}}(\gamma)$, we take a "dual view." Rather than directly defining the margin outliers, we first define the margin inliers, the subsets of $S$ that are $\gamma$-separable. The margin outliers are then simply the complement of the margin inliers.

**Definition 6.1** (Margin inliers/outliers)**.** We define the mar-

gin inliers / outliers of the dataset $S$ with respect to the margin value $\gamma \in [0,1]$ as follows:

$$\mathcal{S}_{\text{in}}(\gamma) := \{S' \subset S \mid \gamma(S') \geq \gamma\}$$
$$\mathcal{S}_{\text{out}}(\gamma) := \{S \setminus S' \mid S' \in \mathcal{S}_{\text{in}}(\gamma)\}$$

This characterization enables us to divide the dataset into two subsets and analyze the optimization error for each separately, as explained in the following section.

### 6.1. Inlier-outlier Analysis of Noisy Gradient Descent

We provide an overview of our inlier-outlier analysis procedure. For a fixed $\gamma \in [0,1]$, we can appropriately decrease the hinge loss parameter $c$ so that the minimum empirical $c$-hinge loss on $S_{\text{in}}(\gamma)$ is reduced to zero:

$$\hat{L}_c(\mathbf{w}; S) \leq \hat{L}_c(\mathbf{w}^*; S) + \text{EER}^{\ 1}$$
$$\leq \underbrace{\hat{L}_c(\mathring{\mathbf{w}}_{\text{in}}; S_{\text{in}}(\gamma))}_{\text{(a) zero if } c \leq \gamma} + \underbrace{\hat{L}_c(\mathring{\mathbf{w}}_{\text{in}}; S_{\text{out}}(\gamma))}_{\text{(b)} \leq \|\mathbf{x}\| \cdot |S_{\text{out}}(\gamma)|/c} + \text{EER}$$
$$\leq \mathcal{O}(|S_{\text{out}}(\gamma)|/c) + \text{EER}$$

(2)

It is worth noting that $\mathbf{w}$ can be generated by any optimization algorithm, offering the flexibility to replace our full-batch gradient descent (Algo. 5) with alternative methods.

As shown in Eq. 2, there is a trade-off between achieving a larger margin and removing fewer data points. As the size of $S_{\text{out}}(\gamma)$ increases, the margin $\gamma(S \setminus S_{\text{out}})$ also increases. However, removing more points makes the remaining dataset less representative of the original distribution, leading to a high error on $S_{\text{out}}(\gamma)$. In an extreme case, for any non-degenerate binary classification problem, at most $n - 2$ points can be removed to make the remaining data linearly separable. However, adapting to such a margin is not meaningful. Since Eq. 2 holds for any $S_{\text{in}} \in \mathcal{S}_{\text{in}}(\gamma)$, we aim to minimize $|S_{\text{out}}(\gamma)|$, ensuring that only the smallest number of "outliers" are removed.

In the next two lemmas, we show that the analysis procedure of Eq. 2 remains valid for random projection based gradient descent. We begin by presenting a margin preservation lemma, leveraging the data-oblivious property of Johnson-Lindenstrauss projections (Larsen & Nelson, 2017).

**Lemma 6.2** (Margin preservation after random projection). *Let $S = \{(\mathbf{x}_i, y_i)\}_{i=1}^n \subset \mathcal{B}^d(1) \times \{\pm 1\}$ and $\gamma \in [0,1]$. Construct a JL projection matrix $\Phi \in \mathbb{R}^{k \times d}$ with entries i.i.d. as $\frac{1}{\sqrt{k}}\text{Rad}(\frac{1}{2})$ where $k = \mathcal{O}(\frac{\log((n+1)(n+2)/\beta)}{\gamma^2})$. For any $S_{\text{in}} \in \mathcal{S}_{\text{in}}(\gamma)$, we have[2]:*

$$\mathbb{P}_\Phi\left(\gamma(\Phi S_{\text{in}}) \geq \frac{\gamma}{3}\right) \geq 1 - \beta$$

---

[1] EER stands for Excess Empirical Risk (Definition A.11); $\mathring{\mathbf{w}}_{\text{in}}$ denotes the normalized max-margin separator for $S_{\text{in}}(\gamma)$; $\mathbf{w}^* = \arg\min_{\mathbf{w} \in \mathbb{R}^d} \hat{L}_c(\mathbf{w}; S)$

[2] $\text{Rad}(\frac{1}{2})$ denotes the Rademacher distribution

The proof is included in Appendix B. Since the margin $\gamma$ is preserved for the largest margin inlier set in $\mathcal{S}_{\text{in}}(\gamma)$, we can formulate the following utility lemma for Algo. 1:

**Lemma 6.3.** *Given GDP budget $\mu > 0$, failure probability $\beta \in (0,1)$, and $\gamma \in (0,1)$, running $\mathcal{A}_{\text{JLGD}}$ (Algo. 1) with $\Phi$ from Lemma 6.2 and hinge loss parameter $\gamma/3$ is $\mu$-GDP. Further, w.p. at least $1 - 2\beta$, the last-iterate estimator $\Phi^\top \mathbf{w} \in \mathbb{R}^d$ satisfies:*

$$\tilde{L}_{\gamma/3}(\Phi^\top \mathbf{w}; S) \leq \min_{S_{\text{out}} \in \mathcal{S}_{\text{out}}(\gamma)} \tilde{\mathcal{O}}\left(\frac{1}{\gamma^2 \mu n} + \frac{|S_{\text{out}}|}{\gamma n}\right)$$

Lemma 6.3 indicates that Algo. 1 adapts to the smallest margin outlier set for a given margin level. We defer the proof to Appendix D.2.

Utility bounds for noisy gradient methods often include a $\sqrt{d}$ term (Bassily et al., 2014). In contrast, the bound provided in Lemma 6.3 is dimension-independent. Compared to existing utility bounds for private gradient descent methods that leverage JL projections, such as Bassily et al. (2022, Lemma 3.1) and results from Arora et al. (2022, Appendix A.4), our Lemma 6.3 avoids a strict $\tilde{\mathcal{O}}(n^{-1/2})$ dependence.

### 6.2. Adapting to the Optimal Margin

Lemma 6.3 suggests that a larger $\gamma$ and a smaller $|S_{\text{out}}|$ lead to a tighter upper bound on the empirical risk. However, increasing the margin $\gamma$ simultaneously causes the size of $S_{\text{out}}$ to grow. To minimize this upper bound, it is necessary to determine the optimal $\gamma \in [0,1]$.

We approach this as a hyperparameter tuning task. First, we discretize the margin range to create a set of candidate hyperparameters. Then, we evaluate each hyperparameter privately and select the one that minimizes the empirical zero-one risk.

Specifically, we construct a doubling grid over $[0,1]$, defined as $\{1/n, 2/n, 4/n, \ldots, 2^{\lfloor \log_2 n \rfloor}/n, 1\}$, and then apply the $\mathcal{A}_{\text{Iter}}$(Algo. 2) using the empirical zero-one loss as the score function. By leveraging the monotonicity of margin outliers, the doubling grid guarantees an acceptable approximation guarantee, as demonstrated in the following lemma:

**Lemma 6.4.** *Given dataset $S \in (\mathcal{B}^d(1) \times \{\pm 1\})^n$ and $\varepsilon > 0$, for doubling set $\Gamma = \{1/n, 2/n, 4/n, ..., 2^{\lfloor \log_2 n \rfloor}/n, 1\}$:*

$$\min_{\substack{\gamma \in \Gamma \\ S_{\text{out}} \in \mathcal{S}_{\text{out}}(\gamma)}} \left(\frac{|S_{\text{out}}|}{n\gamma} + \frac{1}{n\gamma^2 \varepsilon}\right) \wedge 1$$

$$\leq \min_{\substack{S_{\text{out}} \subset S \\ \gamma := \gamma(S \setminus S_{\text{out}}) > 0}} \mathcal{O}\left(\frac{|S_{\text{out}}|}{n\gamma} + \frac{1}{n\gamma^2 \varepsilon}\right) \wedge 1$$

We now state the empirical risk bound for Algo. 3:

**Theorem 6.5** (Empirical risk bound for $\mathcal{M}^*$). *Running $\mathcal{M}^*$ with $\tilde{\mathcal{R}}_S$ satisfies $(\varepsilon, \delta)$-DP, for $\delta \in (0,1)$ and $\varepsilon \in$*

$(0, \ 8\log(1/\delta))$. *In addition,*
*(1) For the averaged estimator $\bar{\mathbf{w}}_{\text{out}} \in \mathbb{R}^d$, we have utility guarantee in expectation:*

$$\mathbb{E}[\tilde{\mathcal{R}}_S(\bar{\mathbf{w}}_{\text{out}})] \leq \min_{\substack{S_{\text{out}} \subset S \\ \gamma := \gamma(S \setminus S_{\text{out}}) > 0}} \tilde{\mathcal{O}}\left(\frac{1}{\gamma^2 n\varepsilon} + \frac{|S_{\text{out}}|}{\gamma n} + \frac{1}{n^2}\right) \wedge 1$$

*(2) For the last-iterate estimator $\mathbf{w}_{\text{out}} \in \mathbb{R}^d$, w.p. at least $1 - 3/n^2$:*

$$\tilde{\mathcal{R}}_S(\mathbf{w}_{\text{out}}) \leq \min_{\substack{S_{\text{out}} \subset S \\ \gamma := \gamma(S \setminus S_{\text{out}}) > 0}} \tilde{\mathcal{O}}\left(\frac{1}{\gamma^2 n\varepsilon} + \frac{|S_{\text{out}}|}{\gamma n} + \frac{1}{n}\right) \wedge 1$$

We defer the proof details of Lemma 6.4 and Theorem 6.5 to Appendix G and Appendix H.

### 6.3. Proof for the Population Risk

In this section, we show a population risk bound for $\mathcal{M}^*$. We begin by presenting an upper bound on the population risk, derived by applying the AM-GM inequality to refine Bassily et al. (2022, Lemma A.1):

**Lemma 6.6.** *Let $S$ be a training set containing $n$ data points sampled i.i.d. from the distribution $D$, and let the hypothesis class defined as $\mathcal{H}_k = \{\mathbf{x} \mapsto \text{sign}(\langle \mathbf{x}, \mathbf{w} \rangle) \mid \mathbf{x} \in \mathbb{R}^k\}$. For any $\mathbf{w}_k \in \mathcal{H}_k$, the following holds w.p. at least $1 - \beta$ over the randomness of sampling:*

$$\mathcal{R}_D(\mathbf{w}_k) \leq 2\tilde{\mathcal{R}}_S(\mathbf{w}_k) + \frac{5(\text{VC}(\mathcal{H}_k)\log(2n)) + \log(\beta/4)}{n}$$

Thus, instead of directly using the empirical zero-one loss for selection, as in the previous section, where the goal was to minimize empirical risk, we use the penalized zero-one loss for selection to minimize population loss:

$$\hat{\mathcal{R}}_S(\mathbf{w}_{\text{out}}) + \frac{5}{2}(\text{VC}(\mathcal{H}_k)\log(2n) + \log(\beta/4)) \quad (3)$$

The remainder of the proof for hyperparameter selection follows from a union bound over private hyperparameter selection. Finally, we provide the population risk guarantee for Algo. 3.

**Theorem 6.7.** *Under the same conditions on $\varepsilon$, $\delta$, $\Gamma$, and $n$ as in Theorem 6.5, Algo. 3, using the score defined in Eq. 6.6, satisfies $(\varepsilon, \delta)$-DP. W.h.p. the output last-iterate estimator $\mathbf{w}_{\text{out}} \in \mathbb{R}^d$ satisfies:*

$$\mathcal{R}_D(\mathbf{w}_{\text{out}}) \leq \min_{\substack{S_{\text{out}} \subset S \\ \gamma := \gamma(S \setminus S_{\text{out}}) > 0}} \tilde{\mathcal{O}}\left(\frac{1}{\gamma^2 n \min\{1, \varepsilon\}} + \frac{|S_{\text{out}}|}{\gamma n}\right)$$

When $|S_{\text{out}}| = o(\sqrt{n})$, the population risk in Theorem 6.7 is on the order of $(1/\sqrt{n})$, improving upon Theorem 3.2 in Bassily et al. (2022), where the bound is $\tilde{\mathcal{O}}(1/\sqrt{n})$. In the realizable case, when data has margin $\gamma$ with probability 1, Theorem 6.7 also recovers the $\tilde{\mathcal{O}}(1/n)$ rate. We defer the proof details to Appendix I.

## 7. Advanced Hyperparameter Tuning

In practice, practitioners often tune multiple hyperparameters, with each of which follows an exponential grid. Iterating over all parameter configurations (as in Algo. 2) would significantly degrade the privacy guarantee. This motivates us to consider an advanced private hyperparameter tuning approach ((Papernot & Steinke, 2022), outlined in Algo. 4), where the final privacy budget grows only logarithmically with the number of repetitions. Compared to Algo. 2, Algo. 4 determines number of repetitions from a specific distribution $Q$, implemented here as a geometric distribution ($\text{TNB}_{1,r}$, Appendix A.1).

---

**Algorithm 4:** $\mathcal{A}_{\text{PrivTune}}(\mathcal{M}, \Theta, Q, S, \mu)$

1 **Input:** Base mechanism $\mathcal{M}$, hyperparameter set $\Theta$,
      run time distribution $Q$, dataset $S$,
      GDP budget $\mu$
2 **Choose:** Criterion function $U$ with sensitivity $\Delta$
3 Sample number of runs $K \sim Q$
4 **for** $t = 1, ..., K$ **do**
5      $\theta_t \sim \text{Uniform}(\Theta)$
6      $m_t = \mathcal{M}(\theta_t; S, \frac{\mu}{\sqrt{2}})$
7      $\xi_t \sim \mathcal{N}(0, \frac{2\Delta^2}{\mu^2})$
8      $u_t = U(m_t; S) + \xi_t$
9 $t_{\text{out}} = \arg\min_{t \in \{1, ..., K\}} u_t$
10 **Output:** $(m_{t_{\text{out}}}, \theta_{t_{\text{out}}})$

---

To incorporate Algo. 4 into our main algorithm, one can simply replace $\mathcal{A}_{\text{Iter}}$ with $\mathcal{A}_{\text{PrivTune}}$ in line 8 of Algo. 3 (see Appendix K.1 for the full statement). The following lemma presents the privacy and utility guarantees, with proofs deferred to Appendix K.4.

**Theorem 7.1.** *Running Algorithm 3 with $\mathcal{A}_{\text{PrivTune}}$ as the hyperparameter selector, margin hyperparameter set $\Theta \subset [0, 1]$, $Q = \text{TNB}_{1, \frac{1}{|\Theta|(n^2-1)}}$, and a GDP budget $\mu = \frac{\varepsilon}{6\sqrt{2\log(|\Theta|(n^2-1)/\delta)}}$ ensures $(\varepsilon + \delta, \delta)$-DP, for any $\delta \in (0, 1)$ and $\varepsilon \in (\delta, 24\log(|\Theta|(n^2-1)/\delta))$. For the averaged estimator $\bar{\mathbf{w}}_{\text{out}} \in \mathbb{R}^d$, we achieve a utility guarantee in expectation:*

$$\mathbb{E}[\tilde{\mathcal{R}}_S(\bar{\mathbf{w}}_{\text{out}})] \leq \min_{\substack{\gamma \in \Theta \\ S_{\text{out}} \subset S_{\text{out}}(\gamma)}} \mathcal{O}\left(\frac{\log(|\Theta| n/\delta)}{\gamma^2 n\varepsilon} + \frac{|S_{\text{out}}|}{\gamma n} + \frac{1}{n^2}\right)$$

As observed, the final utility bound scales only logarithmically with the size of the hyperparameter set $\Theta$. In contrast, for the brute force method $\mathcal{A}_{\text{Iter}}$, the utility bound grows as $|\Theta|^{1/2}$ (Eqn. 17). We note that our utility bound for private hyperparameter tuning extends beyond linear classification (Lemma K.4). To further demonstrate the utility improvement, we examine a general hyperparameter tuning problem

*Table 2.* Results on the population zero-one loss (with high probability). Capital letters $X, W$ denote data space and parameter space correspondingly. $\gamma$ is the data-independent *confidence margin* parameter in Row 1 and 2. In Row 3, $\gamma$ denotes the geometric margin of a data subset after $S_{\text{out}}$ is removed. W.L.O.G., we assume the privacy loss $\varepsilon \leq 1$. All results are stated in the agnostic case.

| Source | Constraints | Result |
|---|---|---|
| Bassily et al. (2022, Thm. 3.1) | $\|\mathbf{x}\| \leq b, \quad \|\mathbf{w}\| \leq C$ | $\tilde{\mathcal{O}}\Big(\frac{1}{n} + \frac{C^2 b^2}{\gamma^2 n} + \frac{C^2 b^2}{\gamma^2 n \varepsilon}\Big)$ $+ \min_{\mathbf{w} \in \mathcal{B}^d(C)} \left( \tilde{\mathcal{R}}_S^\gamma(\mathbf{w}) + \tilde{\mathcal{O}}\left( \sqrt{\tilde{\mathcal{R}}_S^\gamma(\mathbf{w}) \left( \frac{C^2 b^2}{n\gamma^2} + \frac{1}{n} \right)} \right) \right)$ |
| Bassily et al. (2022, Thm. 3.2) | $\|\mathbf{x}\| \leq b, \quad \|\mathbf{w}\| \leq C$ | $\tilde{\mathcal{O}}\Big( \frac{1}{n^{1/2}} + \frac{Cb}{\gamma n^{1/2}} + \frac{Cb}{\gamma n^{1/2} \varepsilon^{1/2}} \Big) + \min_{\mathbf{w} \in \mathcal{B}^d(C)} \tilde{L}_\gamma(\mathbf{w}; S)$ |
| Theorem L.2 | $\|\mathbf{x}\| \leq b$ | $\min_{\substack{S_{\text{out}} \subset S \\ \gamma := \text{margin}(S \setminus S_{\text{out}})}} \tilde{\mathcal{O}}\left( \frac{b^2}{n\gamma^2 \varepsilon} + \frac{b|S_{\text{out}}|}{n\gamma} \right)$ |

with $K$ hyperparameters, each associated with a grid of size $m$, yielding $m^K$ total configurations. In this setting, the utility upper bound for $\mathcal{A}_{\text{Iter}}$ scales as $m^{\mathcal{O}(K)}$, whereas for $\mathcal{A}_{\text{PrivTune}}$, it reduces to $\mathcal{O}(Km)$.

### 7.1. Private Hyperparameter Tuning on a Small Set

In our setting, Algo. 2 and Algo. 4 yield upper bounds with the same dependence on $|\Theta|$. This is because the size of the hyperparameter set is only $\log(n)$ and is dominated by other $\text{poly}(n)$ factors from random projection. In terms of runtime, $\mathcal{A}_{\text{Iter}}$ requires just $\mathcal{O}(\log(n))$ repetitions, whereas $\mathcal{A}_{\text{PrivTune}}$ has an expected runtime of $\mathcal{O}(n^2 \log(n))$ repetitions. Since $\mathcal{A}_{\text{PrivTune}}$ employs uniform random selection (line 5 in Algo. 4), it needs more repetitions to encounter the optimal hyperparameter $\theta^*$. To ensure that the probability of failing to select $\theta^*$ remains below $\beta$, i.e., $\mathbb{P}_Q(\theta^* \text{ not selected}) \leq \beta$, the expected number of repetitions cannot be too small.

**Lemma 7.2.** *Suppose $Q \sim \text{TNB}_{1,r}$ and fauilure probability is set to be $\beta$. If $\mathbb{E}[Q] = 1/r \geq \beta/(1-\beta)(|\Theta|-1)$, Algo. 4 ensures $\mathbb{P}_Q(\theta^* \text{ not selected}) \leq \beta$*

We defer the proof to Appendix M.2. To achieve polynomial decay in the failure probability, i.e., $\beta = \mathcal{O}(n^{-\alpha})$, the expected number of repetitions $\mathbb{E}[Q]$ should be larger than $(|\Theta| - 1)(n^\alpha - 1)$, which grows polynomially with $n$. One might wonder whether truncated negative binomial distributions with different parameters could achieve $\mathcal{O}(\log(n))$ repetitions while preserving the same utility guarantee of Theorem 7.1. In Appendix M.3, we further explore this topic and show that using truncated negative binomial distribution may not be feasible.

## 8. Discussion

### 8.1. More General Case

We extend the risk guarantees to any bounded data space, i.e. $\|\mathbf{x}\| \leq b$, showing that this generalization introduces an additional dependency on $b$, scaling proportionally to $1/\gamma$. A comparison of population risk results is presented in Table 2. We defer proofs to Appendix L. Compared with the results in (Bassily et al., 2022), our bound applies even when the parameter space is unbounded. This is because of the reference point in our inlier-outlier convergence analysis of Algo. 1 is the normalized max-margin separator rather than the empirical risk minimizer. (section 6.1)

### 8.2. Further Improvements and Future Directions

An open question is whether the dependence on $\gamma$ in $\frac{|S_{\text{out}}|}{\gamma n}$ can be eliminated for privately proper learning of halfspaces:

**Question 8.1.** *Given an unknown distribution $D$ and $n$ i.i.d. samples drawn from it, does there exist an efficient DP algorithm $\mathcal{A}$ such that with high probability, the following holds:*

$$\tilde{R}_D(\mathcal{A}(S)) \leq \min_{\substack{S_{\text{out}} \subset S \\ \gamma := \gamma(S \setminus S_{\text{out}}) > 0}} \tilde{\mathcal{O}}\left( \frac{1}{\text{poly}(n, \varepsilon, \gamma)} + \frac{|S_{\text{out}}|}{n} \right)$$

The proper agnostic learning of halfspaces in non-private settings is NP-hard (Guruswami & Raghavendra, 2009; Feldman et al., 2006; Daniely, 2016). Therefore, achieving this improvement is unlikely without making certain assumptions about the noise model.

One might consider the Massart halfspace model (Diakonikolas et al., 2019; Chen et al., 2020; Chandrasekaran et al., 2024; Diakonikolas & Zarifis, 2024). However, since our algorithms rely on convex ERM, directly applying our

approach does not achieve a bound of $\frac{|S_{\mathrm{out}}|}{n}$, as stated by the lower bound in Diakonikolas et al. (2019, Theorem 3.1). More recently, (Chandrasekaran et al., 2024; Diakonikolas & Zarifis, 2024) introduce efficient algorithms that attain an excess population risk of $\tilde{\mathcal{O}}\left(n^{-1/2}\gamma^{-2}\right)$. Exploring whether similar guarantees can be extended to the differentially private setting remains an interesting direction for future research.

## Acknowledgments

The work is partially supported by NSF Award #2048091. The authors would like to thank Vasilis Kontonis, Ming Yin, Lydia Zakynthinou, and Shiwei Zeng for helpful discussions and for pointing out related work. E.W. would also like to thank Yingyu Lin for helpful suggestions on writing.

## Impact Statement

This paper presents work whose goal is to advance the field of Machine Learning. There are many potential societal consequences of our work, none of which we feel must be specifically highlighted here.

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

# A. Extended preliminary and auxiliary lemmas

## A.1. Truncated Negative Binomial Distribution

**Definition A.1** (Truncated Negative Binomial Distribution ((Papernot & Steinke, 2022))). Let $r \in (0,1)$, $\eta \in (-1, \infty)$, and $K \sim \mathrm{TNB}_{\eta, r}$ with support on $\in \{1, 2, 3, ...\}$

- If $\eta \neq 0$:

$$\mathbb{P}(K = k) = \frac{(1-r)^k}{r^{-\eta} - 1} \cdot \prod_{l=0}^{k-1} \left( \frac{l + \eta}{l + 1} \right), \qquad \mathbb{E}[K] = \frac{\eta(1-r)}{r(1-r^\eta)}, \qquad \mathrm{PGF}(x) = \frac{(1 - (1-r)x)^{-\eta} - 1}{r^{-\eta} - 1}$$

- If $\eta = 0$:

$$\mathbb{P}(K = k) = \frac{(1-r)^k}{k \cdot \log(1/r)}, \qquad \mathbb{E}[K] = \frac{1/r - 1}{\log(1/r)}, \qquad \mathrm{PGF}(x) = \frac{\log(1 - (1-r)x)}{\log(r)}$$

*Remark* A.2. When $\eta = 1$, $\mathrm{TNB}_{1,r}$ corresponds to geometric distribution. When $\eta = 0$, $\mathrm{TNB}_{0,r}$ corresponds to logarithmic distribution.

**Fact A.3** (Cumulative distribution function for $\mathrm{TNB}_{1,r}$). *For $t \geq 1$, $\mathbb{P}(\mathrm{TNB}_{1,r} \leq t) = 1 - (1-r)^t$*

**Lemma A.4** (Tail bound for $\mathrm{TNB}_{1,r}$). *For $\beta \in (0,1)$, $\mathbb{P}\left( \mathrm{TNB}_{1,r} > \lceil \frac{\log(\beta)}{\log(1-r)} \rceil \right) \leq \beta$*

*Proof.* By Fact A.3, let $(1-r)^t \leq \beta$, we have $t \geq \lceil \frac{\log(\beta)}{\log(1-r)} \rceil$. $\qquad \square$

## A.2. Concentration Ineqialities

**Lemma A.5** (Concentration of maximum of Gaussians). *Let $\xi_1, ..., \xi_n \overset{\mathrm{iid}}{\sim} \mathcal{N}(0, \sigma^2)$, then:*

- *(in expectation):*

$$\mathbb{E}\left[ \max_{i \in [n]} \xi_i \right] \leq \sigma\sqrt{2\log n}$$

- *(with high probability): For any $\beta \in (0,1)$, we have:*

$$\mathbb{P}\left( \max_{t \in [n]} \xi_t \geq \sigma\sqrt{2\log(2n)} + \sqrt{2\log(1/\beta)} \right) \leq \beta$$

**Corollary A.6** (Concentration for range of Gaussians). *Let $\xi_1, ..., \xi_n \overset{\mathrm{iid}}{\sim} \mathcal{N}(0, \sigma^2)$, let $M := \max_{i \in [n]} \xi_i$ and $m := \min_{i \in [n]} \xi_i$ then:*

$$\mathbb{P}\left( M - m \geq 2\sigma\sqrt{2\log(2n)} + 2\sqrt{2\log(2/\beta)} \right) \leq \beta$$

*Proof.* For any $t \geq 0$:

$$\begin{aligned} \mathbb{P}(M - m \geq t) &\leq \mathbb{P}(\{ -\frac{t}{2} < m < M < \frac{t}{2} \}^c) \\ &\leq \mathbb{P}(M \geq \frac{t}{2}) + \mathbb{P}(m \geq -\frac{t}{2}) \\ &= 2\mathbb{P}(M \geq \frac{t}{2}) \end{aligned} \qquad (4)$$

Set $t = 2\sigma\sqrt{2\log(2n)} + 2\sqrt{2\log(2/\beta)}$, we have:

$$2\mathbb{P}(M \geq \frac{t}{2}) = 2\mathbb{P}(M \geq \sigma\sqrt{2\log(2n)} + \sqrt{2\log(2/\beta)}) \leq \beta$$

$\square$

**Lemma A.7** (Upper bound for maximal of random number of Gaussians). *Let $K \sim \mathrm{TNB}_{\eta,r}$ and $\xi_1, ..., \xi_K \overset{\text{iid}}{\sim} \mathcal{N}(0, \sigma^2)$, we have*

$$\mathbb{E}\left[\min_{t \in [K]} \xi_t\right] \leq \sigma \sqrt{2 \log\left(\frac{\eta(1-r)}{r(1-r^\eta)}\right)}$$

*Proof.*

$$
\begin{aligned}
\mathbb{E}_{K, \mathcal{N}^{\otimes K}}\left[\min_{t \in [K]} \xi_t\right] &= \sum_{j=1}^{\infty} \mathbb{E}_{\mathcal{N}^{\otimes j}}\left[\min_{t \in [j]} \xi_t\right] \cdot \mathbb{P}(K = j) \\
&\leq \sum_{j=1}^{\infty} \sigma \sqrt{2 \log(j)} \cdot \mathbb{P}(K = j) \qquad \text{Lemma A.5} \\
&= \sqrt{2} \sigma \mathbb{E}_{K \sim \mathrm{TNB}_{\eta,r}}[\sqrt{\log(K)}] \\
&\leq \sqrt{2} \sigma \left(\sqrt{\log \mathbb{E}_{K \sim \mathrm{TNB}_{\eta,r}}[K]}\right) \qquad \text{Apply Jensen's Inequality twice} \\
&= \sigma \sqrt{2 \log\left(\frac{\eta(1-r)}{r(1-r^\eta)}\right)}
\end{aligned}
$$

(5)

$\square$

*Remark* A.8. If $K \sim \mathrm{TNB}_{1,r}$, then $\mathbb{E}_{K, \mathcal{N}^{\otimes K}}\left[\min_{t \in [K]} \xi_t\right] \leq \sigma \sqrt{2 \log(1/r)}$

### A.3. Other lemmas

**Lemma A.9** (First order condition for convexity). *Let $f : \mathbb{R}^d \to \mathbb{R}$ be a differentiable convex function. The for any $x, y \in D$, we have:*

$$f(\mathbf{y}) - f(\mathbf{x}) \geq \langle \mathbf{g}, \mathbf{y} - \mathbf{x} \rangle$$

*with $\mathbf{g}$ being sub-gradient for $f$ at $\mathbf{x}$*

**Lemma A.10** ($L_2$ sensitivity for the gradient of $c$-hinge loss). *Let $x \in \mathcal{X}$, the $L_2$ sensitivity of $\nabla_{\mathbf{w}} \hat{L}_c(\mathbf{w}; S) = \sum_{(\mathbf{x}, y) \in S} \nabla_{\mathbf{w}} \ell_c(y\langle \mathbf{w}, \mathbf{x} \rangle)$ is*

*(a) (adding/removing):* $\max_{x \in \mathcal{X}} \frac{\|\mathbf{x}\|}{c}$

*(b) (replacement):* $2 \max_{x \in \mathcal{X}} \frac{\|\mathbf{x}\|}{c}$

*Proof of Lemma A.10.* Lipschitz constant of $\ell_c(y\langle \cdot, \mathbf{x} \rangle)$ is upper bounded by $\|\mathbf{x}\|/c$. $\square$

### A.4. Other definitions

**Definition A.11** (Excess Empirical Risk). *Given dataset $S \in \mathcal{X}^*$ and loss function $L : \mathcal{W} \times \mathcal{X}^* \to \mathbb{R}$, the excess empitical risk for parameter $w \in \mathcal{W}$ is defined as follow:*

$$\mathrm{EER}(w) = L(w; S) - \min_{w \in \mathcal{W}} L(w; S)$$

(6)

## B. Proofs of Margin preservation Lemma 5.1 and Lemma 6.2

The distance and angle preservation lemma presented here is derived from the renowned Johnson-Lindenstrauss lemma (Johnson et al., 1986):

**Lemma B.1** (Lemma A.3 in Bassily et al. (2022)). *Let $X := \{\mathbf{x}_1, ..., \mathbf{x}_n\} \subset \mathbb{R}^d$, distortion rate $e \in (0, 1]$, and $\Phi$ be $k \times d$ random matrix with $\Phi_{ij} \overset{\text{iid}}{\sim} \frac{1}{\sqrt{k}} \mathrm{Rad}(\frac{1}{2})$, with $k = \mathcal{O}\left(\frac{\log(n(n+1)/\beta)}{e^2}\right)$. Then w.p. at least $1 - \beta$ over the randomness of $\Phi$, the following two statements hold simultaneously for any $\mathbf{u}, \mathbf{v} \in X$:*

*(a)* $\sqrt{1 - \frac{e}{3}}\|\mathbf{u}\| \leq \|\Phi\mathbf{u}\| \leq \sqrt{1 + \frac{e}{3}}\|\mathbf{u}\|$

*(b)* $|\langle\Phi\mathbf{u}, \Phi\mathbf{v}\rangle - \langle\mathbf{u}, \mathbf{v}\rangle| \leq \frac{e}{3}\|\mathbf{u}\|\|\mathbf{v}\|$

## B.1. Proof of Lemma 5.1

A formal version of Lemma 5.1 is provided below:

**Lemma B.2** (Formal version of Lemma 5.1). *Let* $S = \{(\mathbf{x}_i, y_i)\}_{i=1}^n \subset \mathbb{R}^d \times \{\pm 1\}$ *with* $\|\mathbf{x}\| \leq b$, *distortion rate parameter* $e \in (0, 1)$, $\Phi \sim \Pi := (\frac{1}{\sqrt{k}}\mathrm{Rad}(\frac{1}{2}))^{k \times d}$ *with* $k = \mathcal{O}\left(\frac{\log((n+1)(n+2)/\beta)}{e^2}\right)$. *Then, for any* $S' \in 2^S$ *with* $\gamma(S') \geq \gamma$, *we have*

$$\mathbb{P}_{\Phi\sim\Pi}\left(\gamma(\Phi S') \geq \frac{\gamma}{2} - \frac{eb}{6}\right) \geq 1 - \beta$$

*Proof.* For any $S' \subset 2^S$ with $\gamma(S') \geq \gamma$, there exist some *unit vector* $\mathbf{w}_{S'} \in \mathbb{R}^d$ s.t. $\gamma(\mathbf{w}_{S'}; S') \geq \gamma$, running JL projection over $S \cup \{\mathbf{w}_{S'}\}$ and $k = \mathcal{O}\left(\frac{\log((n+2)(n+1)/\beta)}{e^2}\right)$ yields the following guarantee by Lemma B.1 (b):

$$\forall \mathbf{x} \in S', \quad \mathbb{P}_{\Phi\sim\Pi}\left(|\langle\Phi\mathbf{w}_{S'}, \Phi\mathbf{x}\rangle - \langle\mathbf{w}_{S'}, \mathbf{x}\rangle| \leq \frac{e}{3}\|\mathbf{w}_{S'}\|\|\mathbf{x}\|\right) \geq 1 - \beta$$

This implies the following happens w.p. at least $1 - \beta$ over the randomness of $\Phi$:

$$\min_{(\mathbf{x},y)\in S'} y\langle\Phi\mathbf{w}_{S'}, \Phi\mathbf{x}\rangle \geq \min_{(\mathbf{x},y)\in S'} y\langle\mathbf{w}_{S'}, \mathbf{x}\rangle - \frac{eb}{3} = \gamma - \frac{eb}{3}$$

Under the same event, with norm preservation stated in Lemma B.1 (a), we get

$$\|\Phi\mathbf{w}_{S'}\| \leq \sqrt{1 + \frac{e}{3}} \cdot \|\mathbf{w}_{S'}\| = \sqrt{1 + \frac{e}{3}}$$

Combing things all together, we get a lower bound for $\gamma(\Phi S')$:

$$\min_{(\mathbf{x},y)\in S} \frac{y\langle\Phi\mathbf{w}_{S'}, \Phi\mathbf{x}\rangle}{\|\Phi\mathbf{w}_{S'}\|} \geq \frac{\gamma - \frac{eb}{3}}{\sqrt{1 + \frac{e}{3}}} \geq \frac{\gamma}{2} - \frac{eb}{6}$$

where the last inequality is by $e \in [0, 1]$. Thus,

$$\mathbb{P}_{\Phi\sim\Pi}\left(\gamma(\Phi S') \geq \frac{\gamma}{2} - \frac{eb}{6}\right) \geq 1 - \beta$$

$\square$

## B.2. Proof of Lemma 6.2

The proof of Lemma 6.2 is by directly applying Lemma B.2

**Lemma B.3** (Restate of Lemma 6.2). *Let* $S = \{(\mathbf{x}_i, y_i)\}_{i=1}^n \subset \mathcal{B}^d(1) \times \{\pm 1\}$ *and* $\gamma \in [0, 1]$. *Construct JL projection matrix* $\Phi \in \mathbb{R}^{k \times d}$ *with entry i.i.d.* $\frac{1}{\sqrt{k}}\mathrm{Rad}(\frac{1}{2})$ *and* $k = \mathcal{O}(\frac{\log((n+1)(n+2)/\beta)}{\gamma^2})$. *For any* $S_{\mathrm{in}} \in \mathcal{S}_{\mathrm{in}}(\gamma)$, *we have*[3]:

$$\mathbb{P}_{\Phi}\left(\gamma(\Phi S_{\mathrm{in}}) \geq \frac{\gamma}{3}\right) \geq 1 - \beta$$

*Proof.* Recall that

$$\mathcal{S}_{\mathrm{in}}(\gamma) := \{S' \subset S : \gamma(S') \geq \gamma\}$$

Thus, for any $S_\gamma^* \in \underset{S' \in \mathcal{S}_{\mathrm{in}}(\gamma)}{\arg\max} |S'|$, by Lemma B.2 with distortion rate $e = \gamma$, we have:

$$\mathbb{P}_{\Phi\sim\Pi}\left(\gamma(\Phi S_\gamma^*) \geq \frac{\gamma}{2} - \frac{\gamma}{6}\right) = \mathbb{P}_{\Phi\sim\Pi}\left(\gamma(\Phi S_\gamma^*) \geq \frac{\gamma}{3}\right) \geq 1 - \beta$$

$\square$

---

[3] $\mathrm{Rad}(\frac{1}{2})$ denotes the Rademacher distribution

# C. Technical Lemmas for noisy gradient descent (Algo. 5)

## C.1. Noisy Gradient Descent

---

**Algorithm 5:** $\mathcal{A}_{\mathsf{NGD}}(\ell(\cdot), S, \mu)$: Noisy Gradient Descent (Bassily et al., 2014)

---

1 **Input** : Data $S = \{\mathbf{x}_i, y_i\}_{i=1}^n \subset \mathbb{R}^d \times \{\pm 1\}$, loss function $\ell(\cdot)$, GDP budget $\mu$

2 **Choose** : some reference point $\mathbf{w}_{\mathrm{ref}} \in \mathbb{R}^d$, initialization $\mathbf{w}_0 \in \mathbb{R}^d$

3 **Set** : $\hat{L}(\mathbf{w}; S) = \sum_{(\mathbf{x}, y) \in S} \ell(\mathbf{w}; (x_i, y_i))$ and $L_2$ sensitivity of $\hat{L}$ is $\Delta$,

 number of iteration $T = n^2 \mu^2$,

 noise parameter $\sigma = \frac{\Delta \sqrt{T}}{\mu} = n \Delta$,

 learning rate $\eta = \sqrt{\frac{\|\mathbf{w}_{\mathrm{ref}} - \mathbf{w}_0\|^2}{T(n^2 \Delta^2 + d\sigma^2)}}$

4 **for** $t = 0, ..., T - 1$ **do**

5     $\xi_t \sim \mathcal{N}(0, \sigma^2 \mathbb{I}_d)$

6     $\mathbf{w}_{t+1} = \mathbf{w}_t - \eta(\nabla_\mathbf{w} \hat{L}(\mathbf{w}_t; S) + \xi_t)$

7 **Output** : Averaged estimator: $\bar{\mathbf{w}} = \frac{1}{T} \sum_{t=0}^{T-1} \mathbf{w}_t$ or Last-iterate estimator: $\mathbf{w} = \mathbf{w}_T$

---

## C.2. Convergence in expectation

**Lemma C.1.** *Algo. 5 is $\mu$-GDP. Furthermore, for the averaged estimator $\bar{\mathbf{w}}$:*

$$\mathbb{E}_\mathcal{A}\left[\hat{L}(\bar{\mathbf{w}}; S)\right] - \hat{L}(\mathbf{w}_{\mathrm{ref}}; S) \leq \mathcal{O}\left(\frac{\|\mathbf{w}_{\mathrm{ref}} - \mathbf{w}_0\| \Delta \sqrt{d}}{\mu}\right) \tag{7}$$

*Proof of Lemma C.1.* The privacy guarantee comes from GDP composition. For notational convenience, denote $g_t := \nabla_\mathbf{w} \hat{L}(\mathbf{w}_t; S)$, $\hat{g}_t = g_t + \xi_t$, then $\mathbf{w}_{t+1} = \mathbf{w}_t - \eta \hat{g}_t$. we have:

$$\|\mathbf{w}_{t+1} - \mathbf{w}_{\mathrm{ref}}\|^2 = \|\mathbf{w}_t - \eta \hat{g}_t - \mathbf{w}_{\mathrm{ref}}\|^2$$
$$= \|\mathbf{w}_t - \mathbf{w}_{\mathrm{ref}}\|^2 - 2\eta \langle \hat{g}_t, \mathbf{w}_t - \mathbf{w}_{\mathrm{ref}} \rangle + \eta^2 \|\hat{g}_t\|^2$$

Taking expectation conditioned on $\mathbf{w}_t$, and by $\mathbb{E}[\hat{g}_t | \mathbf{w}_t] = g_t$, $\mathbb{E}[\hat{g}_t^2 | \mathbf{w}_t] \leq n^2 \Delta^2 + d\sigma^2$ we have:

$$\mathbb{E}[\|\mathbf{w}_{t+1} - \mathbf{w}_{\mathrm{ref}}\|^2] = \mathbb{E}\left(\mathbb{E}\left[\|\mathbf{w}_{t+1} - \mathbf{w}_{\mathrm{ref}}\|^2 | \mathbf{w}_t\right]\right)$$
$$= \mathbb{E}\left(\mathbb{E}\left[\|\mathbf{w}_t - \mathbf{w}_{\mathrm{ref}}\|^2 | \mathbf{w}_t\right] - 2\eta \langle \nabla_\mathbf{w} \hat{L}(\mathbf{w}_t; S), \mathbf{w}_t - \mathbf{w}_{\mathrm{ref}} \rangle + \eta^2 \|\hat{g}_t\|^2\right)$$
$$\leq \mathbb{E}[\|\mathbf{w}_t - \mathbf{w}_{\mathrm{ref}}\|^2] - 2\eta(\mathbb{E}[\hat{L}(\mathbf{w}_t; S)] - \hat{L}(\mathbf{w}_{\mathrm{ref}}; S)) + \eta^2(n^2 \Delta^2 + d\sigma^2) \quad \text{Lemma A.9}$$

By telescoping sum and dividing both sides by $T$, we get:

$$\mathbb{E}\left[\hat{L}(\bar{\mathbf{w}}_T; S)\right] - \hat{L}(\mathbf{w}_{\mathrm{ref}}; S) \leq \mathbb{E}\left[\frac{1}{T} \sum_{t=0}^{T-1} \hat{L}(\mathbf{w}_t; S)\right] - \hat{L}(\mathbf{w}_{\mathrm{ref}}; S)$$
$$\leq \frac{\|\mathbf{w}_{\mathrm{ref}} - \mathbf{w}_0\|^2}{T\eta} + \eta(n^2 \Delta^2 + d\sigma^2) - \frac{\|\mathbf{w}_T - \mathbf{w}_{\mathrm{ref}}\|^2}{T}$$
$$\leq \frac{\|\mathbf{w}_{\mathrm{ref}} - \mathbf{w}_0\|^2}{T\eta} + \eta(n^2 \Delta^2 + d\sigma^2)$$

Since $T = n^2 \mu^2$, $\sigma^2 = n^2 \Delta^2$, and choosing $\eta = \sqrt{\frac{\|\mathbf{w}_{\mathrm{ref}} - \mathbf{w}_0\|^2}{T(n^2 L^2 + d\sigma^2)}}$ to meet the equality condition of AM-GM, we get:

$$\mathbb{E}\left[\hat{L}(\bar{\mathbf{w}}_T; S)\right] - \hat{L}(\mathbf{w}_{\mathrm{ref}}; S) \leq 2\sqrt{\frac{\|\mathbf{w}_{\mathrm{ref}} - \mathbf{w}_0\|^2(n^2 \Delta^2 + d\sigma^2)}{T}}$$
$$\leq \frac{2n\Delta \|\mathbf{w}_{\mathrm{ref}} - \mathbf{w}_0\|}{\sqrt{T}} + \frac{2\sqrt{d}\sigma \|\mathbf{w}_{\mathrm{ref}} - \mathbf{w}_0\|}{\sqrt{T}}$$
$$\leq \frac{4\Delta \sqrt{d} \|\mathbf{w}_{\mathrm{ref}} - \mathbf{w}_0\|}{\mu}$$

$\square$

### C.3. Convergence with high probability

**Lemma C.2** (Theorem 3.5 in Liu & Zhou (2024)). *Running Algo. 5 with step size* $\eta = \sqrt{\frac{\|\mathbf{w}_{\mathrm{ref}} - \mathbf{w}_0\|^2}{T(n^2\Delta^2 + d\sigma^2 \log(1/\beta))}}$ *and output last-iterate estimator* $\mathbf{w}$. *With probability at least* $1 - \beta$,

$$\hat{L}(\mathbf{w}; S) - \hat{L}(\mathbf{w}_{\mathrm{ref}}; S) \le \mathcal{O}\left(\frac{\|\mathbf{w}_{\mathrm{ref}} - \mathbf{w}_0\|\Delta\sqrt{d\log(1/\beta)}}{\mu}\right) \tag{8}$$

## D. Margin Inlier-outlier based convergence analysis for $\mathcal{A}_{\mathsf{JLGD}}$ (Algo. 1)

In this section, we derive convergence bound for $\mathcal{A}_{\mathsf{JLGD}}$ (Algo. 1) characterized by margin inlier-outliers.

### D.1. Expected guarantee

We derive the convergence guarantee for $\mathcal{A}_{\mathsf{JLGD}}$ (Algo. 1), with expectation taking over the randomness of the ERM algorithm.

**Lemma D.1** (Expected convergence bound). *Suppose* $S = S_{\mathrm{in}} \sqcup S_{\mathrm{out}} \subset \mathcal{B}^d(b) \times \{\pm 1\}$ *with* $\gamma(S_{\mathrm{in}}) \ge \gamma$. *Run Algo. 1 with hinge loss* $\ell_{\gamma/3}(\cdot)$, *GDP budget* $\mu$, *and dataset* $S$. *The averaged estimator* $\bar{\mathbf{w}} \in \mathbb{R}^k$ *satisfies: w.p. at least* $1 - \beta$ *over the randomness of JL matrix* $\Phi$:

$$\mathbb{E}_{\mathcal{A}}[\tilde{L}_{\gamma/3}(\bar{\mathbf{w}}; \Phi S)] \le \mathcal{O}\left(\frac{b^2 \log^{1/2}((n+2)(n+1)/\beta)}{n\gamma^2\mu} + \frac{|S_{\mathrm{out}}|}{\gamma n}\right)$$

*Proof of Lemma D.1.* Denote the projected dataset $\Phi S := \{(\mathrm{Proj}_{\mathcal{B}^k(2b)}(\Phi\mathbf{x}_i), y_i)\}_{i=1}^n$. Since $\gamma(S_{\mathrm{in}}) \ge \gamma$ by assumption, apply Lemma B.2 with distortion rate $e = \gamma/b$ and $k = \mathcal{O}\left(\frac{\log((n+2)(n+1)/\beta)}{e^2}\right)$, we have $\gamma(\Phi S_{\mathrm{in}}) \ge \frac{\gamma}{3}$ w.p. at least $1 - \beta$. We denote this event by $E_1$. Let $\mathring{\mathbf{w}}_{\mathrm{in}} \in \mathbb{R}^k$ to be the normalized max-margin separator for $\Phi S_{\mathrm{in}}$, we have $\ell_{\gamma/3}(\mathring{\mathbf{w}}_{\mathrm{in}}; (\Phi\mathbf{x}, y)) = 0$ for any $(\Phi\mathbf{x}, y) \in \Phi S_{\mathrm{in}}$ under event $E_1$.

Now, we initialize $\mathcal{A}_{\mathsf{NGD}}$ with hinge loss function $\ell_{\gamma/3}$, dataset $\Phi S$, GDP budget $\mu$, reference point $\mathbf{w}_{\mathrm{ref}} = \mathring{\mathbf{w}}_{\mathrm{in}}$, initialization point $\mathbf{w}_0 = 0$ and learning rate $\eta = \sqrt{\frac{\|\mathbf{w}_{\mathrm{ref}} - \mathbf{w}_0\|^2}{T(n^2\Delta^2 + d\sigma^2)}} = \sqrt{\frac{1}{T(n^2\Delta^2 + d\sigma^2)}}$. Let $\bar{\mathbf{w}}$ be the averaged estimator outputed from $\mathcal{A}_{\mathsf{NGD}}$ (Algo. 5). Conditioned on the event $E_1$, which happens w.p. at least $1 - \beta$ over the randomness of $\Phi$, we have:

$$\mathbb{E}_{\mathcal{A}}[\hat{L}_{\gamma/3}(\bar{\mathbf{w}}; \Phi S)] \le \hat{L}_{\gamma/3}(\mathring{\mathbf{w}}_{\mathrm{in}}; \Phi S) + \frac{4\Delta\sqrt{k}}{\mu} \qquad\qquad \text{Lemma C.1}$$

$$= \hat{L}_{\gamma/3}(\mathring{\mathbf{w}}_{\mathrm{in}}; \Phi S_{\mathrm{in}}) + \hat{L}_{\gamma/3}(\mathring{\mathbf{w}}_{\mathrm{in}}; \Phi S_{\mathrm{out}}) + \frac{4\Delta\sqrt{k}}{\mu}$$

$$= 0 + \hat{L}_{\gamma/3}(\mathring{\mathbf{w}}_{\mathrm{in}}; \Phi S_{\mathrm{out}}) + \frac{4\Delta\sqrt{k}}{\mu} \qquad\qquad \text{under event } E_1$$

Under the same event, we analyze the worst-case guarantee on the margin outlier set $\Phi S_{\mathrm{out}}$:

$$\hat{L}_{\gamma/3}(\mathring{\mathbf{w}}_{\mathrm{in}}; \Phi S_{\mathrm{out}}) \le |S_{\mathrm{out}}|\left(1 + \frac{3}{\gamma}\|\mathring{\mathbf{w}}_{\mathrm{in}}\| \max_{\mathbf{x} \in S_{\mathrm{out}}} \|\Phi\mathbf{x}\|\right)$$

$$\le |S_{\mathrm{out}}|\left(1 + \frac{3b}{\gamma}\sqrt{1 + \frac{\gamma}{3b}}\right) \qquad\qquad \text{Lemma B.1 norm preservation}$$

$$\le \frac{23b|S_{\mathrm{out}}|}{5\gamma}$$

Given that the projection dimension is $k = \mathcal{O}\left(\frac{b^2 \log((n+2)(n+1)/\beta)}{\gamma^2}\right)$ and the $L_2$ sensitivity of $\ell_{\gamma/3}$ is $\Delta \le \frac{12b}{\gamma}$

(Lemma A.10), we have, with probability at least $1 - \beta$ over $\Phi$, the following bound for the averaged loss:

$$\mathbb{E}_{\mathcal{A}}[\hat{L}_{\gamma/3}(\bar{\mathbf{w}}; \Phi S)] \leq \mathcal{O}\left(\frac{b^2 \log^{1/2}((n+2)(n+1)/\beta)}{n\gamma^2\mu} + \frac{b|S_{\text{out}}|}{n\gamma}\right)$$

$\square$

Using the above expected convergence Lemma D.1 and margin preservation Lemma 6.2 we have the following corollary:

**Corollary D.2.** *Given $\gamma \in [0, b]$, running Algo. 1 with hinge loss $\ell_{\gamma/3}(\cdot)$, GDP budget $\mu$, and dataset $S$. The averaged estimator $\bar{\mathbf{w}} \in \mathbb{R}^k$ satisfies: w.p. at least $1 - \beta$ over the randomness of JL matrix $\Phi$:*

$$\mathbb{E}_{\mathcal{A}}[\tilde{L}_{\gamma/3}(\bar{\mathbf{w}}; \Phi S)] \leq \min_{S_{\text{out}} \in \mathcal{S}_{\text{out}}(\gamma)} \mathcal{O}\left(\frac{b^2 \log^{1/2}((n+2)(n+1)/\beta)}{n\gamma^2\mu} + \frac{b|S_{\text{out}}|}{\gamma n}\right)$$

### D.2. High Probability Guarantee, Lemma 6.3

Using Lemma C.2, we can also establish the high-probability version of Lemma D.1.

**Lemma D.3** (High probability convergence bound, Lemma 6.3)**.** *Under the same algorithmic conditions as Corollary D.2, with probability at least $1 - 2\beta$, the last-iterate estimator $\mathbf{w}$ from Algo. 1 satisfies:*

$$\tilde{L}_{\gamma/3}(\mathbf{w}; \Phi S) \leq \min_{S_{\text{out}} \in \mathcal{S}_{\text{out}}(\gamma)} \mathcal{O}\left(\frac{b^2 \log^{1/2}(1/\beta) \log^{1/2}((n+1)(n+2)/\beta)}{n\gamma^2\mu} + \frac{b|S_{\text{out}}|}{\gamma n}\right)$$

*Proof.* Let the projected dataset be denoted as $\Phi S := \{(\text{Proj}_{\mathcal{B}^k(2b)}(\Phi\mathbf{x}_i), y_i)\}_{i=1}^n$. Given the assumption that $\gamma(S_{\text{in}}) \geq \gamma$, we apply Lemma B.2 with a distortion rate $e = \gamma/b$ and $k = \mathcal{O}\left(\frac{\log((n+2)(n+1)/\beta)}{e^2}\right)$. This ensures that $\gamma(\Phi S_{\text{in}}) \geq \frac{\gamma}{3}$ with probability at least $1 - \beta$. We denote this event as $E_1$. Let $\mathring{\mathbf{w}}_{\text{in}} \in \mathbb{R}^k$ be the normalized max-margin separator for $\Phi S_{\text{in}}$. Under event $E_1$, we have $\ell_{\gamma/3}(\mathring{\mathbf{w}}_{\text{in}}; (\Phi\mathbf{x}, y)) = 0$ for any $(\Phi\mathbf{x}, y) \in \Phi S_{\text{in}}$.

We now initialize $\mathcal{A}_{\text{NGD}}$ with the loss function $\ell = \ell_{\gamma/3}$, dataset $\Phi S$, GDP budget $\mu$, reference point $\mathbf{w}_{\text{ref}} = \mathring{\mathbf{w}}_{\text{in}}$, initialization point $\mathbf{w}_0 = 0$, and learning rate $\eta = \sqrt{\frac{\|\mathbf{w}_{\text{ref}} - \mathbf{w}_0\|^2}{T(n^2\Delta^2 + d\sigma^2 \log(1/\beta))}} = \sqrt{\frac{1}{T(n^2\Delta^2 + d\sigma^2 \log(1/\beta))}}$. Let $\mathbf{w}$ denote the last-iterate estimator obtained from $\mathcal{A}_{\text{NGD}}$ (Algo. 5). Conditioned on the margin-preserving event and the successful execution of $\mathcal{A}_{\text{NGD}}$, which occurs with probability at least $1 - 2\beta$, we have:

$$\hat{L}_{\gamma/3}(\mathbf{w}; \Phi S) \leq \mathcal{O}\left(\hat{L}_{\gamma/3}(\mathring{\mathbf{w}}_{\text{in}}; \Phi S_{\text{in}}) + \hat{L}_{\gamma/3}(\mathring{\mathbf{w}}_{\text{in}}; \Phi S_{\text{out}}) + \frac{\Delta\sqrt{k\log(1/\beta)}}{\mu}\right) \quad \text{Lemma C.2}$$

$$= \mathcal{O}\left(\hat{L}_{\gamma/3}(\mathring{\mathbf{w}}_{\text{in}}; \Phi S_{\text{out}}) + \frac{\Delta\sqrt{k\log(1/\beta)}}{\mu}\right) \quad \text{under margin preservation event } E_1$$

$$\leq \mathcal{O}\left(\frac{b|S_{\text{out}}|}{\gamma} + \frac{b\sqrt{\log(1/\beta)}}{\mu\gamma} \cdot \frac{b\sqrt{\log((n+1)(n+2)/\beta)}}{\gamma}\right)$$

$$= \mathcal{O}\left(\frac{b|S_{\text{out}}|}{\gamma} + \frac{b^2\sqrt{\log(1/\beta)\log((n+1)(n+2)/\beta)}}{\mu\gamma^2}\right)$$

Finally, dividing both sides by $n$ and applying margin preservation Lemma 6.2 we obtain, with probability at least $1 - 2\beta$:

$$\tilde{L}_{\gamma/3}(\mathbf{w}; \Phi S) \leq \min_{S_{\text{out}} \in \mathcal{S}_{\text{out}}(\gamma)} \mathcal{O}\left(\frac{b^2 \log^{1/2}(1/\beta) \log^{1/2}((n+1)(n+2)/\beta)}{n\mu\gamma^2} + \frac{b|S_{\text{out}}|}{\gamma n}\right)$$

$\square$

# E. Privacy Analysis for $\mathcal{A}_{\mathsf{Iter}}$ (Algo. 2)

### E.1. Technical lemmas

We first present a conversion lemma from Gaussian DP to approximate DP:

**Lemma E.1** (GDP to approximate DP conversion). *Suppose mechanism $\mathcal{M}$ satisfies $\mu$-GDP, then $\mathcal{M}$ is also $(\varepsilon, \delta)$-DP with*

$$\varepsilon \le \frac{\mu^2}{2} + \mu\sqrt{2\log(1/\delta)}$$

*Proof.* By the definition of Gaussian Differential Privacy, we know that $\mathcal{M}$ is dominated by the Gaussian mechanism with a sensitivity of $1$ and a noise parameter $\sigma = 1/\mu$, i.e.

$$H_{e^\varepsilon}(\mathcal{M}(X)\|\mathcal{M}(X')) \le H_{e^\varepsilon}(\mathcal{N}(0,1)\|\mathcal{N}(\mu,1)) = H_{e^\varepsilon}(\mathcal{N}(0,1/\mu^2)\|\mathcal{N}(1,1/\mu^2))$$

The rest of the proof follows the same procedure as in the proof of Lemma 5.5 in (Koskela et al., 2024), with $\varepsilon$ set to be $\frac{\mu^2}{2} + \mu\sqrt{2\log(1/\delta)}$. $\square$

Under a high privacy regime, we have a cleaner upper bound for $\varepsilon$:

**Corollary E.2.** *Under the same condition as Lemma E.1, for any fixed $\delta > 0$ if further assume $\mu \le 2\sqrt{2\log(1/\delta)}$, then $\mathcal{M}$ satisfies $\left(2\mu\sqrt{2\log(1/\delta)}, \delta\right)$-DP.*

### E.2. Privacy analysis for $\mathcal{A}_{\mathsf{Iter}}$

It's equivalent to analyzing the privacy guarantee for compositions between GDP mechanisms:

**Lemma E.3** (Privacy for $\mathcal{A}_{\mathsf{Iter}}$ (Algo. 2)). *Given any $\delta > 0$ and $\varepsilon \le 8\log(1/\delta)$, let $\{\mathcal{M}_i\}_{i=1}^m$ being a sequence of mechanisms with $\mathcal{M}_i$ satisfing $\frac{\varepsilon}{2\sqrt{2m\log(1/\delta)}}$-GDP. The composed mechanism $\mathcal{M} := \mathcal{M}_m \circ \ldots \circ \mathcal{M}_1$ satisfies $(\varepsilon, \delta)$-DP*

*Proof.* By adaptive composition of Gaussian DP, we have: $\mathcal{M}$ satisfies $\frac{\varepsilon}{2\sqrt{2\log(1/\delta)}}$-GDP. Thus, applying Corollary E.2, we have $\mathcal{M}$ satisfies $(\varepsilon, \delta)$-DP $\square$

# F. Utility analysis for $\mathcal{A}_{\mathsf{Iter}}$ (Algo. 2)

In this section, we consider a generalized version of $\mathcal{A}_{\mathsf{Iter}}$. Namely, we have noisy observations $\{U_\theta + \xi_\theta\}_{\theta \in \Theta}$, and $\theta_{\mathrm{out}} = \arg\min_{\theta \in \Theta}\{U_\theta + \xi_\theta\}$, where $\xi_\theta \overset{\mathrm{iid}}{\sim} N(0, \sigma^2)$. We are interested in bounding the deviation between $U_{\theta_{\mathrm{out}}}$ and $\min_{\theta \in \Theta} U_\theta$.

### F.1. Expected guarantee

**Lemma F.1** (Expected guarantee). *For possibly random $\{U_\theta\}_{\theta \in \Theta}$, we have:*

$$\mathbb{E}_\xi[U_{\theta_{\mathrm{out}}}] \le \min_{\theta \in \Theta} U_\theta + 2\sigma\sqrt{2\log(|\Theta|)}$$

*where the expectation is taken over the randomness of Gaussian noise*

*Proof.*

$$\min_{\theta \in \Theta}\{U_\theta + \xi_\theta\} = U_{\theta_{\mathrm{out}}} + \xi_{\theta_{\mathrm{out}}} \qquad \text{definition of } \theta_{\mathrm{out}}$$

$$\ge U_{\theta_{\mathrm{out}}} + \min_{\theta \in \Theta} \xi_\theta$$

which implies:

$$U_{\theta_{\mathrm{out}}} \le \min_{\theta \in \Theta}\{U_\theta + \xi_\theta\} - \min_{\theta \in \Theta} \xi_\theta \tag{9}$$

On the other hand:

$$\min_{\theta \in \Theta}\{U_\theta + \xi_\theta\} \le \min_{\theta \in \Theta} U_\theta + \max_{\theta \in \Theta} \xi_\theta$$

Putting things together, we have:

$$U_{\theta_{\text{out}}} \le \min_{\theta \in \Theta} U_\theta + \max_{\theta \in \Theta} \xi_\theta - \min_{\theta \in \Theta} \xi_\theta$$

Finally, by taking expectation w.r.t. Gaussian noises, we have:

$$\mathbb{E}[U_{\theta_{\text{out}}}] \le \min_{\theta \in \Theta} U_\theta + \mathbb{E}[\max_{\theta \in \Theta} \xi_\theta] - \mathbb{E}[\min_{\theta \in \Theta} \xi_\theta]$$

By $\xi$ is symmetric, we have:

$$\mathbb{E}[\min_{\theta \in \Theta} \xi_\theta] = -\mathbb{E}[\max_{\theta \in \Theta} -\xi_\theta] = -\mathbb{E}[\max_{\theta \in \Theta} \xi_\theta]$$

Thus,

$$\mathbb{E}[U_{\theta_{\text{out}}}] \le \min_{\theta \in \Theta} U_\theta + 2\mathbb{E}[\max_{\theta \in \Theta} \xi_\theta]$$
$$\le \min_{\theta \in \Theta} U_\theta + 2\sigma\sqrt{2\log(|\Theta|)}$$

where the last inequality is by Lemma A.5. □

### F.2. High probability guarantee

**Lemma F.2** (High probability guarantee). *For possibly random $\{U_\theta\}_{\theta \in \Theta}$, w.p. at least $1 - \beta$ over the randomness of Gaussians:*

$$U_{\theta_{\text{out}}} \le \min_{\theta \in \Theta} U_\theta + 2\sigma\sqrt{2\log(2|\Theta|)} + 2\sqrt{2\log(2/\beta)}$$

*where the expectation is taken over the randomness of Gaussian noise*

*Proof.* Similarly, as the proof in the previous Lemma, we have:

$$U_{\theta_{\text{out}}} \le \min_{\theta \in \Theta} U_\theta + \max_{\theta \in \Theta} \xi_\theta - \min_{\theta \in \Theta} \xi_\theta$$

The remaining proof is by applying Lemma A.6. □

## G. Proof of Lemma 6.4

**Lemma G.1.** *Given dataset $S \in (\mathcal{B}^d(1) \times \{\pm 1\})^n$, for any $\gamma \in [1/n, 1]$, there exist $\tilde{\gamma}$ from doubling set $\Gamma = \{1/n, 2/n, 4/n, ..., 1\}$, such that:*

$$\min_{S_{\text{out}} \in \mathcal{S}_{\text{out}}(\tilde{\gamma})} \left( \frac{|S_{\text{out}}|}{n\tilde{\gamma}} + \frac{1}{n\tilde{\gamma}^2\varepsilon} \right) \le \min_{S_{\text{out}} \in \mathcal{S}_{\text{out}}(\gamma)} 4 \left( \frac{|S_{\text{out}}|}{n\gamma} + \frac{1}{n\gamma^2\varepsilon} \right) \tag{10}$$

*Proof.* For any $S_{\text{out}} \in \mathcal{S}_{\text{out}}(\gamma)$ with $\gamma \ge \frac{1}{n}$, by the property of the doubling set $\Gamma$ from $1/n$ to $1$, there always exists $\tilde{\gamma} \in \Gamma$ such that $\gamma/2 \le \tilde{\gamma} \le \gamma$. Also, for any $S_{\text{out}} \in \mathcal{S}_{\text{out}}(\gamma)$, observe that:

$$|S_{\text{out}}| \ge \min_{\tilde{S} \in \mathcal{S}_{\text{out}}(\gamma)} |\tilde{S}| \ge \min_{\tilde{S} \in \mathcal{S}_{\text{out}}(\tilde{\gamma})} |\tilde{S}|$$

The two facts together imply that:

$$\begin{aligned}
\min_{S_{\text{out}} \in \mathcal{S}_{\text{out}}(\tilde{\gamma})} \left( \frac{|S_{\text{out}}|}{n\tilde{\gamma}} + \frac{1}{n\tilde{\gamma}^2\varepsilon} \right) &\le \min_{S_{\text{out}} \in \mathcal{S}_{\text{out}}(\tilde{\gamma})} \left( \frac{2|S_{\text{out}}|}{n\gamma} + \frac{4}{n\gamma^2\varepsilon} \right) \\
&\le \min_{S_{\text{out}} \in \mathcal{S}_{\text{out}}(\gamma)} 4 \left( \frac{|S_{\text{out}}|}{n\gamma} + \frac{1}{n\gamma^2\varepsilon} \right)
\end{aligned} \tag{11}$$

Taking minimum on both sides implies:

$$\min_{\substack{\tilde{\gamma} \in \Gamma \\ S_{\text{out}} \in \mathcal{S}_{\text{out}}(\tilde{\gamma})}} \left( \frac{|S_{\text{out}}|}{n\tilde{\gamma}} + \frac{1}{n\tilde{\gamma}^2\varepsilon} \right) \le \min_{\substack{\gamma \in [1/n, 1] \\ S_{\text{out}} \in \mathcal{S}_{\text{out}}(\gamma)}} 4 \left( \frac{|S_{\text{out}}|}{n\gamma} + \frac{1}{n\gamma^2\varepsilon} \right) \tag{12}$$

□

### G.1. Proof of Lemma 6.4

*Proof of Lemma 6.4.* W.L.O.G., we assme $\varepsilon \leq n$. $\gamma < 1/n$ implies $\frac{1}{n\gamma^2\varepsilon} > 1$. Together with Lemma G.1, we have:

$$
\min_{\substack{\gamma \in \Gamma \\ S_{\text{out}} \in \mathcal{S}_{\text{out}}(\gamma)}} \left( \frac{|S_{\text{out}}|}{n\gamma} + \frac{1}{n\gamma^2\varepsilon} \right) \wedge 1 \leq \min_{\substack{\gamma \in (0,1] \\ S_{\text{out}} \in \mathcal{S}_{\text{out}}(\gamma)}} \mathcal{O}\left( \frac{|S_{\text{out}}|}{n\gamma} + \frac{1}{n\gamma^2\varepsilon} \right) \wedge 1
$$

$$
= \min_{\substack{S_{\text{out}} \subset S \\ \gamma := \gamma(S \setminus S_{\text{out}}) > 0}} \mathcal{O}\left( \frac{|S_{\text{out}}|}{n\gamma} + \frac{1}{n\gamma^2\varepsilon} \right) \wedge 1
$$

(13)

$\square$

We note that Lemma 6.4 remains valid even when the data are not confined to the unit ball:

**Corollary G.2.** *Given dataset $S \in (\mathcal{B}^d(b) \times \{\pm 1\})^n$, and doubling set $\Gamma = \{b/n, 2b/n, 4b/n, ..., b\}$, we have:*

$$
\min_{\substack{\gamma \in \Gamma \\ S_{\text{out}} \in \mathcal{S}_{\text{out}}(\gamma)}} \left( \frac{|S_{\text{out}}|}{n\gamma} + \frac{1}{n\gamma^2\varepsilon} \right) \wedge 1 \leq \min_{\substack{S_{\text{out}} \subset S \\ \gamma := \gamma(S \setminus S_{\text{out}}) > 0}} \mathcal{O}\left( \frac{|S_{\text{out}}|}{n\gamma} + \frac{1}{n\gamma^2\varepsilon} \right) \wedge 1
$$

(14)

## H. Proof of Theorem 6.5, empirical risk bound for Algo. 3

**Theorem H.1** (Theorem 6.5 restated). *Running $\mathcal{M}^*$ with $\tilde{\mathcal{R}}_S$ satisfies $(\varepsilon, \delta)$-DP, for $\delta \in (0, 1)$ and $\varepsilon \in (0, \ 8\log(1/\delta))$. In addition,*
*(1) For the averaged estimator $\bar{\mathbf{w}}_{\text{out}} \in \mathbb{R}^d$, we have utility guarantee in expectation:*

$$
\mathbb{E}[\tilde{\mathcal{R}}_S(\bar{\mathbf{w}}_{\text{out}})] \leq \min_{\substack{S_{\text{out}} \subset S \\ \gamma := \gamma(S \setminus S_{\text{out}}) > 0}} \tilde{\mathcal{O}}\left( \frac{1}{\gamma^2 n\varepsilon} + \frac{|S_{\text{out}}|}{\gamma n} + \frac{1}{n^2} \right) \wedge 1
$$

*(2) For the last-iterate estimator $\mathbf{w}_{\text{out}} \in \mathbb{R}^d$, w.p. at least $1 - 3/n^2$:*

$$
\tilde{\mathcal{R}}_S(\mathbf{w}_{\text{out}}) \leq \min_{\substack{S_{\text{out}} \subset S \\ \gamma := \gamma(S \setminus S_{\text{out}}) > 0}} \tilde{\mathcal{O}}\left( \frac{1}{\gamma^2 n\varepsilon} + \frac{|S_{\text{out}}|}{\gamma n} + \frac{1}{n} \right) \wedge 1
$$

The privacy analysis is the composition of GDP mechanisms, which directly follows Lemma E.3.

### H.1. Proof of Theorem 6.5, in expectation version

We first introduce the following utility lemma:

**Lemma H.2** (Expected ERM). *Under the condition of Theorem 6.5, let $\bar{\mathbf{w}}_{\text{out}} \in \mathbb{R}^d$ be the average estimator output from Algo. 3, we have:*

$$
\mathbb{E}_{\mathcal{A}}[\tilde{\mathcal{R}}_S(\bar{\mathbf{w}}_{\text{out}})] = \min_{\gamma \in \Gamma} \min_{S_{\text{out}} \in \mathcal{S}_{\text{out}}(\gamma)} \tilde{\mathcal{O}}\left( \frac{b^2}{n\gamma^2\varepsilon} + \frac{b|S_{\text{out}}|}{\gamma n} + \frac{1}{n\varepsilon} + \frac{1}{n^2} \right)
$$

*Proof.* Since $\mathcal{A}_{\text{Iter}}$ (Algo. 2) allocates privacy budget evenly, let's denote $\mu$ to be the GDP budget for each call of $\mathcal{A}_{\text{JLGD}}$(Algo. 1).

By Lemma D.2, the following convergence guarantee holds, w.p. at least $1 - \beta$ over the randomness of JL projection, for running $\mathcal{A}_{\text{JLGD}}$ with margin loss $\ell_{\gamma/3}$:

$$
\mathbb{E}_{\mathcal{A}}[\tilde{L}_{\gamma/3}(\bar{\mathbf{w}}_{\gamma}; \Phi_{\gamma}S)] \leq \min_{S_{\text{out}} \in \mathcal{S}_{\text{out}}(\gamma)} \mathcal{O}\left( \frac{b^2 \log^{1/2}((n+2)(n+1)/\beta)}{n\gamma^2\mu} + \frac{b|S_{\text{out}}|}{\gamma n} \right)
$$

(15)

Since the additive Gaussian noise has distribution $\mathcal{N}(0, 1/\mu^2)$, by Lemma F.1 the expected guarantee of $\mathcal{A}_{\text{Iter}}$ is:

$$
\begin{aligned}
\mathbb{E}_{\mathcal{A}}[\tilde{\mathcal{R}}_S(\bar{\mathbf{w}}_{\text{out}})] &\leq \min_{\gamma \in \Gamma} \mathbb{E}_{\mathcal{A}}[\tilde{\mathcal{R}}_S(\Phi_\gamma^\top \bar{\mathbf{w}}_\gamma)] + \frac{2\sqrt{2\log(|\Gamma|)}}{n\mu} \\
&\leq \min_{\gamma \in \Gamma} \mathbb{E}_{\mathcal{A}}[\tilde{L}_{\gamma/3}(\bar{\mathbf{w}}_\gamma; \Phi_\gamma S)] + \frac{2\sqrt{2\log(|\Gamma|)}}{n\mu} \\
&\leq \min_{\gamma \in \Gamma} \min_{S_{\text{out}} \in \mathcal{S}_{\text{out}}(\gamma)} \mathcal{O}\left( \frac{b^2 \log^{1/2}((n+2)(n+1)/\beta)}{n\gamma^2\mu} + \frac{b|S_{\text{out}}|}{\gamma n} \right) + \frac{2\sqrt{2\log(|\Gamma|)}}{n\mu}
\end{aligned}
$$

where the second inequality is by hinge loss upper bounds zero-one loss, and the third inequality is by Eq. 15. Thus, w.p. $1 - |\Gamma|\beta$ over the randomness of JL projection, we have:

$$
\mathbb{E}_{\mathcal{A}}[\tilde{\mathcal{R}}_S(\bar{\mathbf{w}}_{\text{out}})] \leq \min_{\gamma \in \Gamma} \min_{S_{\text{out}} \in \mathcal{S}_{\text{out}}(\gamma)} \underbrace{\mathcal{O}\left( \frac{b^2 \log^{1/2}((n+2)(n+1)/\beta)}{n\gamma^2\mu} + \frac{b|S_{\text{out}}|}{\gamma n} \right)}_{(a)} + \underbrace{\frac{2\sqrt{2\log(|\Gamma|)}}{n\mu}}_{(b)}
$$

Plug in $|\Gamma| = \lceil \log_2(n) \rceil + 1$, $\mu = \frac{\varepsilon}{4\sqrt{|\Gamma|\log(1/\delta)}}$, and scale $\beta$ by $1/|\Gamma|$, we have:

$$
\begin{aligned}
(a) &= \left( \frac{b^2 \log^{1/2}((n+2)(n+1)/\beta)}{n\gamma^2\mu} + \frac{b|S_{\text{out}}|}{\gamma n} \right) \\
&= \left( \frac{4b^2 |\Gamma|^{1/2} \log^{1/2}(|\Gamma|(n+2)(n+1)/\beta) \log^{1/2}(1/\delta)}{n\gamma^2\varepsilon} + \frac{b|S_{\text{out}}|}{\gamma n} \right) \\
&\leq \left( \frac{8b^2 \log_2^{1/2}(n) \log^{1/2}(4(n+2)(n+1)\log_2(n)/\beta) \log^{1/2}(1/\delta)}{n\gamma^2\varepsilon} + \frac{b|S_{\text{out}}|}{\gamma n} \right)
\end{aligned}
\tag{16}
$$

where the last inequality holds when $n \geq 2$, we have $|\Gamma| = \lceil \log_2(n) \rceil + 1 \leq 4\log_2(n)$

$$
\begin{aligned}
(b) &= \frac{2\sqrt{2\log(|\Gamma|)}}{n\mu} \\
&= \frac{8\sqrt{2}|\Gamma|^{1/2} \log^{1/2}(1/\delta) \log^{1/2}(|\Gamma|)}{n\varepsilon} \\
&\leq \frac{16\sqrt{2} \log_2^{1/2}(n) \log^{1/2}(1/\delta) \log^{1/2}(4\log_2(n))}{n\varepsilon}
\end{aligned}
\tag{17}
$$

Setting $\beta = 1/n^2$ and noticing that zero-one loss is upper bounded by 1, we have:

$$
\begin{aligned}
\mathbb{E}_{\mathcal{A}}[\tilde{\mathcal{R}}_S(\Phi_{\text{out}}^\top \bar{\mathbf{w}}_{\text{out}})] &\leq \min_{\gamma \in \Gamma} \min_{S_{\text{out}} \in \mathcal{S}_{\text{out}}(\gamma)} \mathcal{O}\left( \frac{8b^2 \log_2^{1/2}(n) \log^{1/2}(4n^2(n+2)(n+1)\log_2(n)) \log^{1/2}(1/\delta)}{n\gamma^2\varepsilon} + \frac{b|S_{\text{out}}|}{\gamma n} \right) \\
&\quad + \frac{16\sqrt{2} \log_2^{1/2}(n) \log^{1/2}(1/\delta) \log^{1/2}(4\log_2(n))}{n\varepsilon} + \frac{1}{n^2} \\
&= \min_{\gamma \in \Gamma} \min_{S_{\text{out}} \in \mathcal{S}_{\text{out}}(\gamma)} \tilde{\mathcal{O}}\left( \frac{b^2}{n\gamma^2\varepsilon} + \frac{b|S_{\text{out}}|}{\gamma n} + \frac{1}{n\varepsilon} + \frac{1}{n^2} \right)
\end{aligned}
$$

$\square$

Now, we start the proof of Theorem 6.5, in expectation version:

*Proof.* By Lemma H.2, we have:

$$\mathbb{E}_{\mathcal{A}}[\tilde{\mathcal{R}}_S(\bar{\mathbf{w}}_{\text{out}})] \leq \min_{\gamma \in \Gamma} \min_{S_{\text{out}} \in \mathcal{S}_{\text{out}}(\gamma)} \tilde{\mathcal{O}}\left(\frac{1}{n\gamma^2\varepsilon} + \frac{|S_{\text{out}}|}{\gamma n} + \frac{1}{n\varepsilon} + \frac{1}{n^2}\right)$$

Since $\tilde{\mathcal{R}}_S(\bar{\mathbf{w}}_{\text{out}})$ is upper bounded by 1, we have

$$\mathbb{E}_{\mathcal{A}}[\tilde{\mathcal{R}}_S(\bar{\mathbf{w}}_{\text{out}})] \leq \min_{\gamma \in \Gamma} \min_{S_{\text{out}} \in \mathcal{S}_{\text{out}}(\gamma)} \tilde{\mathcal{O}}\left(\frac{1}{n\gamma^2\varepsilon} + \frac{|S_{\text{out}}|}{\gamma n} + \frac{1}{n^2}\right) \wedge 1$$

$$= \min_{\substack{S_{\text{out}} \subset S \\ \gamma := \gamma(S \setminus S_{\text{out}}) > 0}} \tilde{\mathcal{O}}\left(\frac{1}{\gamma^2 n\varepsilon} + \frac{|S_{\text{out}}|}{\gamma n} + \frac{1}{n^2}\right) \wedge 1$$

where the first inequality is by $\gamma < 1$ and the second line is by Lemma 6.4 $\qquad\square$

## H.2. Proof of Theorem 6.5, high probability version

**Lemma H.3** (High probability ERM). *Under the condition of Theorem 6.5, let $\mathbf{w}_{\text{out}} \in \mathbb{R}^d$ be the last-iterate estimator output from Algo. 3, we have w.p. at least $1 - 3/n^2$:*

$$\tilde{\mathcal{R}}_S(\mathbf{w}_{\text{out}}) \leq \min_{\gamma \in \Gamma} \min_{S_{\text{out}} \in \mathcal{S}_{\text{out}}(\gamma)} \tilde{\mathcal{O}}\left(\frac{b^2}{n\gamma^2\varepsilon} + \frac{b|S_{\text{out}}|}{\gamma n} + \frac{1}{n\varepsilon} + \frac{1}{n}\right)$$

*Proof.* By Lemma D.3, for a single call of $\mathcal{A}_{\text{JLGD}}$ that optimizing $\ell_{\gamma/3}$, w.p. at least $1 - 2\beta/|\Gamma|$, the last-iterate estimator $\mathbf{w}_\gamma$ satisfies:

$$\tilde{L}_{\gamma/3}(\mathbf{w}_\gamma; \Phi S) \leq \min_{S_{\text{out}} \in \mathcal{S}_{\text{out}}(\gamma)} \mathcal{O}\left(\frac{b^2 \log^{1/2}(|\Gamma|/\beta) \log^{1/2}(|\Gamma|(n+1)(n+2)/\beta)}{n\gamma^2\mu} + \frac{b|S_{\text{out}}|}{\gamma n}\right)$$

Similarly as the proof for Lemma F.1 and notice that additive gaussian noise has distribution $\mathcal{N}(0, 1/\mu^2)$, and by union bound over $|\Gamma|$ calls of $\mathcal{A}_{\text{JLGD}}$ we have w.p. $1 - 3\beta$:

$$\tilde{\mathcal{R}}_S(\mathbf{w}_{\text{out}}) \leq \min_{\gamma \in \Gamma} \tilde{\mathcal{R}}_S(\Phi_\gamma^\top \mathbf{w}_\gamma) + \frac{2\sqrt{2\log(2|\Gamma|)}}{n\mu} + \frac{2\sqrt{2\log(2/\beta)}}{n}$$

$$\leq \min_{\gamma \in \Gamma} \tilde{L}_{\gamma/3}(\Phi_\gamma^\top \mathbf{w}_\gamma; S) + \frac{2\sqrt{2\log(2|\Gamma|)}}{n\mu} + \frac{2\sqrt{2\log(2/\beta)}}{n}$$

$$\leq \min_{\gamma \in \Gamma} \min_{S_{\text{out}} \in \mathcal{S}_{\text{out}}(\gamma)} \mathcal{O}\underbrace{\left(\frac{b^2 \log^{1/2}(|\Gamma|/\beta) \log^{1/2}(|\Gamma|(n+1)(n+2)/\beta)}{n\gamma^2\mu} + \frac{b|S_{\text{out}}|}{\gamma n}\right)}_{(a)}$$

$$+ \underbrace{\frac{2\sqrt{2\log(2|\Gamma|)}}{n\mu}}_{(b)} + \underbrace{\frac{2\sqrt{2\log(2/\beta)}}{n}}_{(c)}$$

Notice that $|\Gamma| = \lceil\log_2(n)\rceil + 1 \leq 4\log_2(n)$, $\mu = \frac{\varepsilon}{4\sqrt{|\Gamma|\log(1/\delta)}}$, we have:

$$(a) = \frac{b^2 \log^{1/2}(|\Gamma|/\beta) \log^{1/2}(|\Gamma|(n+1)(n+2)/\beta)}{n\gamma^2\mu} + \frac{b|S_{\text{out}}|}{\gamma n}$$

$$\leq \frac{8b^2 \log^{1/2}(4\log_2(n)/\beta) \log^{1/2}(4(n+1)(n+2)\log_2(n)/\beta) \log_2^{1/2}(n) \log^{1/2}(1/\delta)}{n\gamma^2\mu} + \frac{b|S_{\text{out}}|}{\gamma n}$$

$$(b) = \frac{2\sqrt{2\log(2|\Gamma|)}}{n\mu} \leq \frac{16\sqrt{2}\log^{1/2}(8\log_2(n)) \log_2^{1/2}(n) \log^{1/2}(1/\delta)}{n\varepsilon}$$

Putting everything together and set $\beta = 1/n^2$, we have:

$$\tilde{\mathcal{R}}_S(\mathbf{w}_{\text{out}})$$

$$\leq \min_{\gamma \in \Gamma} \min_{S_{\text{out}} \in \mathcal{S}_{\text{out}}(\gamma)} \mathcal{O}\left( \frac{b^2 \log^{1/2}(4\log_2(n)/\beta) \log^{1/2}(4(n+1)(n+2)\log_2(n)/\beta) \log_2^{1/2}(n) \log^{1/2}(1/\delta)}{n\gamma^2\mu} + \frac{b|S_{\text{out}}|}{\gamma n} \right)$$

$$+ \frac{16\sqrt{2}\log^{1/2}(8\log_2(n)) \log_2^{1/2}(n) \log^{1/2}(1/\delta)}{n\varepsilon}$$

$$+ \frac{2\sqrt{2\log(2/\beta)}}{n}$$

$$= \min_{\gamma \in \Gamma} \min_{S_{\text{out}} \in \mathcal{S}_{\text{out}}(\gamma)} \tilde{\mathcal{O}}\left( \frac{b^2}{n\gamma^2\varepsilon} + \frac{b|S_{\text{out}}|}{\gamma n} + \frac{1}{n\varepsilon} + \frac{1}{n} \right)$$

$\square$

Now, we start the proof of Theorem 6.5, high probability version:

*Proof.* By Lemma H.3, w.p. at least $1 - 3/n^2$:

$$\tilde{\mathcal{R}}_S(\mathbf{w}_{\text{out}}) \leq \min_{\gamma \in \Gamma} \min_{S_{\text{out}} \in \mathcal{S}_{\text{out}}(\gamma)} \tilde{\mathcal{O}}\left( \frac{1}{n\gamma^2\varepsilon} + \frac{|S_{\text{out}}|}{\gamma n} + \frac{1}{n\varepsilon} + \frac{1}{n} \right)$$

Since $\tilde{\mathcal{R}}_S(\bar{\mathbf{w}}_{\text{out}})$ is upper bounded by 1, $\gamma < 1$, and applying Lemma 6.4, we have

$$\tilde{\mathcal{R}}_S(\mathbf{w}_{\text{out}}) \leq \min_{\substack{S_{\text{out}} \subset S \\ \gamma := \gamma(S \backslash S_{\text{out}}) > 0}} \tilde{\mathcal{O}}\left( \frac{1}{\gamma^2 n\varepsilon} + \frac{|S_{\text{out}}|}{\gamma n} + \frac{1}{n} \right) \wedge 1$$

$\square$

## I. Proofs for population error bound, Theorem 6.7

**Theorem I.1** (Restate of theorem 6.7). *Under the same conditions for $\varepsilon$, $\delta$, $\Gamma$, and $n$ as in Theorem 6.5, Algo. 3 using score defined in Eq. 6.6 satisfies $(\varepsilon, \delta)$-DP. W.p. $1 - 4/n^2$, the output last-iterate estimator $\mathbf{w}_{\text{out}} \in \mathbb{R}^d$ satisfies:*

$$\mathcal{R}_D(\mathbf{w}_{\text{out}}) \leq \min_{\substack{S_{\text{out}} \subset S \\ \gamma := \gamma(S \backslash S_{\text{out}}) > 0}} \tilde{\mathcal{O}}\left( \frac{1}{\gamma^2 n \min\{1, \varepsilon\}} + \frac{|S_{\text{out}}|}{\gamma n} \right)$$

### I.1. Proof of Lemma 6.6

We first state the relative deviation bound:

**Lemma I.2** (Lemma A.1 in Bassily et al. (2022)). *$S \sim D^{\otimes n}$. For any hypothesis set $\mathcal{H}$ of functions mapping from $\mathcal{X}$ to $\mathbb{R}$, with probability at least $1 - \beta$ over the randomness of samples, the following inequality holds for all $h \in \mathcal{H}$:*

$$\mathcal{R}_D(h) \leq \tilde{\mathcal{R}}_S(h) + 2\sqrt{\tilde{\mathcal{R}}_S(h)\frac{\text{VC}(\mathcal{H})\log(2n) + \log(4/\beta)}{n}} + 4\frac{\text{VC}(\mathcal{H})\log(2n) + \log(4/\beta)}{n}$$

Using this Lemma, we can prove Lemma 6.6:

*Proof of Lemma 6.6.* By AM-GM inequality,

$$2\sqrt{\tilde{\mathcal{R}}_S(h)\frac{\text{VC}(\mathcal{H})\log(2n) + \log(4/\beta)}{n}} \leq \tilde{\mathcal{R}}_S(h) + \frac{\text{VC}(\mathcal{H})\log(2n) + \log(4/\beta)}{n}$$

Thus by Lemma I.2:

$$\mathcal{R}_D(h) \leq 2\tilde{\mathcal{R}}_S(h) + 5\frac{\text{VC}(\mathcal{H})\log(2n) + \log(4/\beta)}{n}$$

$\square$

### I.2. Proof of Theorem 6.7

*Proof.* For any $\gamma \in \Gamma$, since $k_\gamma = \mathcal{O}\left(\frac{\log(|\Gamma|(n+2)(n+1)/\beta)}{\gamma^2}\right)$. Denote $\mathrm{score}(\gamma) := \tilde{\mathcal{R}}_S(w_{k_\gamma}) + \frac{5(\mathrm{VC}(\mathcal{H}_{k_\gamma})\log(2n)+\log(\beta/4))}{n}$.
For privacy analysis, we notice that under the replacement neighboring relationship, $\Delta(\mathrm{score}(\gamma)) = 1$. With the same privacy parameter setting as in Theorem 6.5, we have the algorithm follows $(\varepsilon, \delta)$-DP.

Now we begin utility analysis. In the beginning, we state the uniform convergence theorem for projected data:

For function class $\mathcal{H}_{k_\gamma} := \{\mathbf{x} \mapsto \mathrm{sign}(\langle \mathbf{x}, \mathbf{w}\rangle) \mid \mathbf{x} \in \mathbb{R}^{k_\gamma}\}$, by uniform convergence, we notice that w.p. $1-\beta$ over the randomness of sampling from $D$, for any $\mathbf{w}_\gamma \in \mathbb{R}^{k_\gamma}$:

$$\mathcal{R}_{\Phi D}(\mathbf{w}_\gamma) \leq 2\tilde{\mathcal{R}}_{\Phi S}(\mathbf{w}_\gamma) + \frac{5}{n}(\mathrm{VC}(\mathcal{H}_{k_\gamma})\log(2n) + \log(4/\beta))$$

By Lemma F.2 and uniform convergence, w.p. at least $1 - 2\beta$:

$$\mathcal{R}_D(\mathbf{w}_{\mathrm{out}}) \leq \min_{\gamma \in \Gamma}\left(2\tilde{\mathcal{R}}_S(\Phi_\gamma^\top \mathbf{w}_\gamma) + \underbrace{\frac{5}{n}(\mathrm{VC}(\mathcal{H}_\gamma)\log(2n) + \log(4|\Gamma|/\beta))}_{(a)}\right) + \underbrace{\frac{2\sqrt{2\log(2|\Gamma|)}}{\mu n}}_{(b)} + \underbrace{\frac{2\sqrt{2\log(2/\beta)}}{n}}_{(c)} \quad (18)$$

Since $\beta = 1/n^2$, Notice that $k_\gamma = \mathcal{O}\left(\frac{\log\left(\frac{|\Gamma|(n+2)(n+1)}{\beta}\right)}{\gamma^2}\right)$, $|\Gamma| \leq 4\log_2(n)$ when $n > 2$, we have $\mathrm{VC}(\mathcal{H}_k) \leq \mathcal{O}\left(\frac{\log(4n^2(n+2)(n+1)\log_2(n))}{\gamma^2}\right)$.

Also, $\frac{1}{\mu} = \frac{4\sqrt{|\Gamma|\log(1/\delta)}}{\varepsilon} = \frac{8\sqrt{\log_2(n)\log(1/\delta)}}{\varepsilon}$, we have:

$$(a) = \frac{5}{n}(\mathrm{VC}(\mathcal{H}_\gamma)\log(2n) + \log(4|\Gamma|/\beta))$$
$$\leq \frac{5\log(4n^2(n+2)(n+1)\log_2(n))\log(2n)}{n\gamma^2} + \frac{5\log(16n^2\log_2(n))}{n} = \tilde{\mathcal{O}}\left(\frac{1}{n\gamma^2} + \frac{1}{n}\right)$$
$$(b) = \frac{2\sqrt{2\log(2|\Gamma|)}}{\mu n} \leq \frac{16\sqrt{2}\log_2^{1/2}(n)\log^{1/2}(1/\delta)\log^{1/2}(8\log_2(n))}{n\varepsilon} = \tilde{\mathcal{O}}\left(\frac{1}{n\varepsilon}\right)$$
$$(c) = \frac{2\sqrt{2\log(2/\beta)}}{n} = \frac{4\log^{1/2}(2n)}{n} = \tilde{\mathcal{O}}\left(\frac{1}{n}\right)$$

Thus,

$$\mathcal{R}_D(\mathbf{w}_{\mathrm{out}}) \leq \min_{\gamma \in \Gamma} \tilde{\mathcal{O}}\left(\tilde{\mathcal{R}}_S(\Phi_\gamma^\top \mathbf{w}_\gamma) + \frac{1}{n\gamma^2}\right) + \tilde{\mathcal{O}}\left(\frac{1}{n} + \frac{1}{n\varepsilon}\right) \quad (19)$$

By zero-one loss is upper bounded by hinge loss and applying Lemma 6.3:

$$\tilde{\mathcal{R}}_S(\Phi_\gamma^\top \mathbf{w}_\gamma) \leq \tilde{L}_{\gamma/3}(\Phi_\gamma^\top \mathbf{w}_\gamma; S) \leq \min_{S_{\mathrm{out}} \in \mathcal{S}_{\mathrm{out}}(\gamma)} \tilde{\mathcal{O}}\left(\frac{1}{n\gamma^2\varepsilon} + \frac{|S_{\mathrm{out}}|}{\gamma n}\right) \quad (20)$$

Putting everything together:

$$\mathcal{R}_D(\mathbf{w}_{\mathrm{out}}) \leq \min_{\substack{\gamma \in \Gamma \\ S_{\mathrm{out}} \in \mathcal{S}_{\mathrm{out}}(\gamma)}} \tilde{\mathcal{O}}\left(\frac{1}{n\gamma^2}(1 + \varepsilon^{-1}) + \frac{|S_{\mathrm{out}}|}{n\gamma}\right) + \tilde{\mathcal{O}}\left(\frac{1}{n} + \frac{1}{n\varepsilon}\right)$$
$$\leq \min_{\substack{\gamma \in \Gamma \\ S_{\mathrm{out}} \in \mathcal{S}_{\mathrm{out}}(\gamma)}} \tilde{\mathcal{O}}\left(\frac{1}{n\gamma^2}(1 + \varepsilon^{-1}) + \frac{|S_{\mathrm{out}}|}{n\gamma}\right) \quad (21)$$
$$\leq \min_{\substack{\gamma \in \Gamma \\ S_{\mathrm{out}} \in \mathcal{S}_{\mathrm{out}}(\gamma)}} \tilde{\mathcal{O}}\left(\frac{1}{n\gamma^2\min\{1,\varepsilon\}} + \frac{|S_{\mathrm{out}}|}{n\gamma}\right)$$

where the second inequality holds by $\gamma \le 1$. Finally, by argument similar to Theorem 6.4 and the fact that $\mathcal{R}_D(\mathbf{w}_{\text{out}}) \le 1$, we have:

$$\mathcal{R}_D(\mathbf{w}_{\text{out}}) \le \min_{\substack{S_{\text{out}} \subset S \\ \gamma(S \setminus S_{\text{out}}) > 0}} \tilde{\mathcal{O}}\left(\frac{1}{n\gamma^2}(1 + \varepsilon^{-1}) + \frac{|S_{\text{out}}|}{n\gamma}\right) \wedge 1 \tag{22}$$

which holds w.p. at least $1 - 4/n^2$. $\qquad\square$

## J. Proof of Theorem 4.1

*Proof of Theorem 4.1.* The privacy guarantee and the empirical risk bound are derived in Theorem 6.5; The population risk bound follows by Theorem 6.7. $\qquad\square$

## K. Proof of Theorem 7.1

### K.1. Alternative algorithm using advanced private hyperparameter tuning

---

**Algorithm 6:** $\mathcal{M}'(S, \varepsilon, \delta)$

---

1 **Input:** dataset $S = \{\mathbf{x}_i, y_i\}_{i=1}^n$, Privacy budget $\varepsilon, \delta$
2 **Set:** Margin grid $\Gamma = \{\frac{1}{n}, \frac{2}{n}, \frac{4}{n}, ..., \frac{2^{\lfloor \log_2 n \rfloor}}{n}, 1\}$, Run time distribution $Q = \text{TNB}_{1, \frac{1}{|\Gamma|(n^2-1)}}$,
   GDP budget $\mu = \frac{\varepsilon}{6\sqrt{2\log(|\Gamma|(n^2-1)/\delta)}}$,
3 Initilize hyperparameter set $\Theta = \phi$
4 **for** $\gamma \in \Gamma$ **do**
5   $k_\gamma = \mathcal{O}\left(\frac{1}{\gamma^2}\log(\frac{|\Gamma|(n+2)(n+1)}{\beta})\right)$
6   $\Phi_\gamma \sim (\text{Rad}(\frac{1}{2})/\sqrt{k_\gamma})^{k_\gamma \times d}$
7   $\Theta = \Theta \cup \{(\gamma, \Phi_\gamma)\}$
8 $(\tilde{\mathbf{w}}_{\text{out}}, \gamma_{\text{out}}, \Phi_{\gamma_{\text{out}}}) = \mathcal{A}_{\text{PrivTune}}(\mathcal{A}_{\text{JLGD}}(\cdot), \Theta, Q, S, \mu)$    *Algo. 4*
9 **Output:** $(\tilde{\mathbf{w}}_{\text{out}}, \gamma_{\text{out}}, \Phi_{\gamma_{\text{out}}})$

---

### K.2. Privacy guarantee for $\mathcal{A}_{\text{PrivTune}}$ (Algo. 4)

**Lemma K.1** (Corollary 5.5 in (Koskela et al., 2024)). *Let run time distribution $Q \sim \text{TNB}_{1,r}$, suppose base mechanism is $\mu$-GDP. Then for fixed $\delta > 0$, the private selection algorithm $\mathcal{A}$ is $(\varepsilon, \delta)$-DP for:*

$$\varepsilon = \frac{3}{2}\mu^2 + 3\mu\sqrt{2\log\left(\frac{1}{r \cdot \delta}\right)} + \delta \tag{23}$$

*where $\mathcal{A}$ is defined in Theorem 2 of Papernot & Steinke (2022))*

In the high privacy regime, we can relax the parameter $\varepsilon$ in the above lemma, obtaining a simplified expression for $\varepsilon$ that depends linearly on $\mu$. This leads to more straightforward privacy accounting for Algo. 4.

**Corollary K.2.** *Conditioned on Lemma K.1, if the GDP budget of base mechanism $\mu \le 2\sqrt{2\log\left(\frac{1}{r\delta}\right)}$, then private selection mechanism $\mathcal{A}$ is also $\left(6\mu\sqrt{2\log\left(\frac{1}{r \cdot \delta}\right)} + \delta, \delta\right)$-DP.*

*Proof.* By Lemma K.1 and the upper bound on $\mu$, we have:

$$\varepsilon = \frac{3}{2}\mu^2 + 3\mu\sqrt{2\log\left(\frac{1}{r \cdot \delta}\right)} + \delta \le 6\mu\sqrt{2\log\left(\frac{1}{r \cdot \delta}\right)} + \delta$$

$\qquad\square$

Now, we begin to prove the GDP guarantee for $\mathcal{A}_{\mathsf{PrivTune}}$ (Algo. 4):

**Theorem K.3.** $\mathcal{A}_{\mathsf{PrivTune}}$ *(Algo. 4) with run time distribution* $Q = \mathrm{TNB}_{1,r}$ *and input GDP* $\mu = \frac{\varepsilon}{6\sqrt{2\log(1/(r\delta))}}$ *satisfies* $(\varepsilon + \delta, \delta)$-*DP, for any* $\varepsilon \leq 24\log(\frac{1}{r\delta})$.

*Proof of Lemma K.3.* For any $\theta \in \Theta$, we view the instantiated "base" mechanism, denoted by $\mathcal{A}_\theta$, as the adaptive composition of two mechanisms:

(1) $\mathcal{M}(\theta; S, \frac{\mu}{\sqrt{2}})$ with GDP budget $\frac{\mu}{\sqrt{2}}$;

(2) Gaussian mechanism with $\sigma = \frac{\Delta^2}{(\mu/\sqrt{2})^2}$, with $\Delta$ being the $L_2$ sensitivity of score function $U$.

By GDP composition, $\mathcal{A}_\theta$ is $\mu$-GDP. The remaining proof follows Corollary K.2. $\qquad\square$

### K.3. Utility guarantee for $\mathcal{A}_{\mathsf{PrivTune}}$ (Algo. 4)

**Lemma K.4** (Utility of $\mathcal{A}_{\mathsf{PrivTune}}$ (Algo. 4), in expectation). *Let* $U_1, ..., U_{|\Theta|}$ *be possibly random score corresponding to parameters in* $\Theta$, $K \sim \mathrm{TNB}_{\eta,r}$, *observe* $\{U_{i_t} + \xi_{i_t}\}_{t \in [K]}$[4], *with* $\xi_{i_t} \overset{\mathrm{iid}}{\sim} \mathcal{N}(0, \sigma^2)$, *let* $U_{\mathrm{out}} = \min_{t \in [K]}(U_{i_t} + \xi_{i_t})$ *(break tie arbitrarily) and* $t_{\mathrm{out}}$ *be corresponding index, then w.p. at least* $1 - \mathrm{PGF}_K(1 - 1/|\Theta|)$:

$$\mathbb{E}[U_{\mathrm{out}}] \leq \min_{\theta \in \Theta} \mathbb{E}[U_\theta] + \sigma\sqrt{2\log\left(\frac{\eta(1-r)}{r(1-r^\eta)}\right)}$$

*Proof.* First, we notice that:

$$\min_{t \in [K]}\{U_{i_t} + \xi_{i_t}\} = U_{\mathrm{out}} + \xi_{t_{\mathrm{out}}} \qquad \text{definition of } \tilde{w}_{\mathrm{out}}$$

$$\geq U_{\mathrm{out}} + \min_{t \in [K]} \xi_{i_t} \quad t_{\mathrm{out}} \text{ belongs to } [K]$$

which implies:

$$U_{\mathrm{out}} \leq \min_{t \in [K]}\{U_{i_t} + \xi_{i_t}\} - \min_{t \in [K]} \xi_{i_t}$$

$$= \min_{t \in [K]}\{U_{i_t} + \xi_{i_t}\} + \max_{t \in [K]} \xi_{i_t} \quad \xi \overset{d}{=} -\xi \tag{24}$$

On the other hand:

$$\min_{t \in [K]}\{U_{i_t} + \xi_{i_t}\} \leq \min_{t \in [K]}\{U_{i_t}\} + \max_{t \in [K]}\{\xi_{i_t}\}$$

Denote $Y := \{\mathrm{index}(\theta^*) \in \{i_j\}_{j \in [K]}\}$, where $\theta^* \in \arg\min_{\theta \in \Theta} \mathbb{E}[L(w_\theta)]$. Conditioned on $Y$, we have:

$$\mathbb{E}[\min_{t \in [K]}\{U_{i_t} + \xi_{i_t}\}|Y] \leq \mathbb{E}[\min_{t \in [K]}\{U_{i_t}\}|Y] + \mathbb{E}[\max_{t \in [K]}\{\xi_{i_t}\}|Y]$$

$$\leq \min_{\theta \in \Theta} \mathbb{E}[L(w_\theta)] + \mathbb{E}[\max_{t \in [K]} \xi_t]$$

The last inequality is by (1) using Jensen's inequality and then by the fact $\theta^* \in \mathcal{I}$ under event $Y$; (2) $\xi_t$ and event $Y$ are independent. Putting things together, we have:

$$\mathbb{E}[U_{\mathrm{out}}|Y] \leq \mathbb{E}[\min_{t \in [K]}\{U_{i_t} + \xi_{it}\}|Y] + \mathbb{E}[\max_{t \in [K]} \xi_{i_t}|Y]$$

$$\leq \min_{\theta \in \Theta} \mathbb{E}[L(w_\theta)] + 2\mathbb{E}[\max_{t \in [K]} \xi_t]$$

$$\leq \min_{\theta \in \Theta} \mathbb{E}[L(w_\theta)] + \sigma\sqrt{2\log\left(\frac{\eta(1-r)}{r(1-r^\eta)}\right)} \quad \text{Lemma A.7}$$

The remaining step is by noticing that $\mathbb{P}(Y) \geq 1 - \mathrm{PGF}_K(1 - 1/|\Theta|)$. $\qquad\square$

---

[4] $\{i_j\}_{j \in [K]}$ is a multi-set of $\{1, 2, ..., |\Theta|\}$

### K.4. Proof of Theorem 7.1

We first prove a utility lemma for non-private hyperparameter selection over $\gamma \in \Gamma$:

**Lemma K.5** (union bound for non-private grid search). $\mu > 0$, *For every* $\gamma \in \Gamma$, *set projection dimension* $k_\gamma = \mathcal{O}\left(\frac{1}{\gamma^2}\log\left(\frac{|\Gamma|(n+2)(n+1)}{\beta}\right)\right)$ *and JL projection matrix* $\Phi_\gamma \in \mathbb{R}^{d \times k_\gamma}$, *running* $\mathcal{A}_{JLGD}(\Phi_\gamma, \ell_{\gamma/3}, S, \mu)$ *to get averaged estimator* $\Phi_\gamma^\top \bar{\mathbf{w}}_\gamma$. *Then the following holds w.p. at least* $1 - \beta$:

$$\min_{\gamma \in \Gamma} \mathbb{E}_{\mathcal{A}}[\tilde{L}_{\gamma/3}(\bar{\mathbf{w}}_\gamma; \Phi_\gamma S)] \leq \min_{\substack{\gamma \in \Gamma \\ S_{\text{out}} \in \mathcal{S}_{\text{out}}(\gamma)}} \mathcal{O}\left(\frac{\log^{1/2}(|\Gamma|(n+2)(n+1)/\beta)}{\gamma^2 n \mu} + \frac{|S_{\text{out}}|}{\gamma n}\right) \wedge 1 \tag{25}$$

*proof of Lemma K.5.* By Lemm 6.3, for any fixed $\gamma$, we have the following holds w.p. at least $1 - \beta/|\Gamma|$ over the randomness of $\Phi_\gamma$:

$$\mathbb{E}_{\mathcal{A}}[\tilde{L}_{\gamma/3}(\bar{\mathbf{w}}_\gamma; \Phi_\gamma S)] \leq \min_{S_{\text{out}} \in \mathcal{S}_{\text{out}}(\gamma)} \mathcal{O}\left(\frac{\log^{1/2}(|\Gamma|(n+2)(n+1)/\beta)}{\gamma^2 n \mu} + \frac{|S_{\text{out}}|}{\gamma n}\right)$$

The remaining proof is by taking union bound over $\Gamma$ and taking minimum over $\gamma$ on both sides of the inequality above. $\square$

**Lemma K.6.** *Let* $\Gamma$ *be any finite subset of* $[0,1]$, $\beta = n^{-2}$, $Q = \text{TNB}_{1, \frac{1}{|\Gamma|(n^2-1)}}$, $\mu = \frac{\varepsilon}{6\sqrt{2\log(|\Gamma|(n^2-1)/\delta)}}$. $\mathcal{M}^*$ *(Algo. 6) invokes with* $S$, $Q$, $\Gamma$ *and* $\mu$ *is* $(\varepsilon + \delta, \delta)$*-DP for any* $\varepsilon \leq 24\log(|\Gamma|(n^2-1)/\delta)$. *In addition, the output averaged estimator* $\bar{\mathbf{w}}_{\text{out}} \in \mathbb{R}^d$ *satisfies: w.p. at least* $1 - 2\beta$,

$$\mathbb{E}[\tilde{\mathcal{R}}_S(\bar{\mathbf{w}}_{\text{out}})] \leq \min_{\substack{\gamma \in \Gamma \\ S_{\text{out}} \in \mathcal{S}_{\text{out}}(\gamma)}} \mathcal{O}\left(\frac{\log^{1/2}(|\Gamma|(n+2)(n+1)/\beta)\log^{1/2}(|\Gamma|(n^2-1)/\delta)}{\gamma^2 \varepsilon n} + \frac{|S_{\text{out}}|}{n\gamma}\right) + \frac{12\log(n^2|\Gamma|/\delta)}{n\varepsilon}$$

*Proof of Theorem K.6.* By Theorem K.3, the $\mathcal{M}^*$ is $(\varepsilon + \delta, \delta)$-DP.

Denote event $Y := \{\gamma^* \text{ being selected}\}$, we get $\mathbb{P}(Y) \geq 1 - \beta$. Under event $Y$:

$$\mathbb{E}[\hat{\mathcal{R}}_S(\bar{\mathbf{w}}_{\text{out}})] \leq \min_{\gamma \in \Gamma} \mathbb{E}[\hat{L}_{\gamma/3}(\bar{\mathbf{w}}_\gamma; \Phi_\gamma S)] + \frac{\sqrt{2}}{\mu}\sqrt{\log(|\Gamma|(n^2-1))} \qquad \text{Lemma K.4}$$

$$= \min_{\gamma \in \Gamma} \mathbb{E}[\hat{L}_{\gamma/3}(\bar{\mathbf{w}}_\gamma; \Phi_\gamma S)] + \frac{12\log^{1/2}(|\Gamma|(n^2-1))\log^{1/2}(|\Gamma|(n^2-1)/\delta)}{\varepsilon}$$

$$\leq \min_{\gamma \in \Gamma} \mathbb{E}[\hat{L}_{\gamma/3}(\bar{\mathbf{w}}_\gamma; \Phi_\gamma S)] + \frac{12\log(n^2|\Gamma|/\delta)}{\varepsilon}$$

$$\leq \min_{\substack{\gamma \in \Gamma \\ S_{\text{out}} \in \mathcal{S}_{\text{out}}(\gamma)}} \mathcal{O}\left(\frac{\log^{1/2}(|\Gamma|(n+2)(n+1)/\beta)}{\mu\gamma^2} + \frac{|S_{\text{out}}|}{\gamma}\right) + \frac{12\log(n^2|\Gamma|/\delta)}{\varepsilon} \qquad \text{Lemma K.5}$$

$$\leq \min_{\substack{\gamma \in \Gamma \\ S_{\text{out}} \in \mathcal{S}_{\text{out}}(\gamma)}} \mathcal{O}\left(\frac{\log^{1/2}(|\Gamma|(n+2)(n+1)/\beta)\log^{1/2}(|\Gamma|(n^2-1)/\delta)}{\gamma^2 \varepsilon} + \frac{|S_{\text{out}}|}{\gamma}\right) + \frac{12\log(n^2|\Gamma|/\delta)}{\varepsilon}$$

Thus, at least $1 - 2\beta$:

$$\mathbb{E}[\tilde{\mathcal{R}}_S(\bar{\mathbf{w}}_{\text{out}})] \leq \min_{\substack{\gamma \in \Gamma \\ S_{\text{out}} \in \mathcal{S}_{\text{out}}(\gamma)}} \mathcal{O}\left(\frac{\log^{1/2}(|\Gamma|(n+2)(n+1)/\beta)\log^{1/2}(|\Gamma|(n^2-1)/\delta)}{\gamma^2 \varepsilon n} + \frac{|S_{\text{out}}|}{n\gamma}\right) + \frac{12\log(n^2|\Gamma|/\delta)}{n\varepsilon}$$

$$\tag{26}$$

$\square$

*Proof of Theorem 7.1.* By Lemma K.6 and $|\Gamma| \leq 4\log_2(n)$ when $n > 2$, we have

$$
\begin{aligned}
\mathbb{E}[\tilde{\mathcal{R}}_S(\bar{\mathbf{w}}_{\text{out}})] &\leq \min_{\substack{\gamma \in \Gamma \\ S_{\text{out}} \in \mathcal{S}_{\text{out}}(\gamma)}} \tilde{\mathcal{O}}\left(\frac{1}{\gamma^2 n\varepsilon} + \frac{|S_{\text{out}}|}{\gamma n}\right) + \frac{12\log(n^2\log_2(n)/\delta)}{n\varepsilon} + 2\beta \\
&= \min_{\substack{\gamma \in \Gamma \\ S_{\text{out}} \in \mathcal{S}_{\text{out}}(\gamma)}} \tilde{\mathcal{O}}\left(\frac{1}{\gamma^2 n\varepsilon} + \frac{|S_{\text{out}}|}{\gamma n}\right) + \frac{12\log(n^2\log_2(n)/\delta)}{n\varepsilon} + \frac{2}{n^2} \\
&= \min_{\substack{\gamma \in \Gamma \\ S_{\text{out}} \in \mathcal{S}_{\text{out}}(\gamma)}} \tilde{\mathcal{O}}\left(\frac{1}{\gamma^2 n\varepsilon} + \frac{|S_{\text{out}}|}{\gamma n}\right) + \tilde{\mathcal{O}}\left(\frac{1}{n\varepsilon} + \frac{1}{n^2}\right) \\
&= \min_{\substack{\gamma \in \Gamma \\ S_{\text{out}} \in \mathcal{S}_{\text{out}}(\gamma)}} \tilde{\mathcal{O}}\left(\frac{1}{\gamma^2 n\varepsilon} + \frac{|S_{\text{out}}|}{\gamma n} + \frac{1}{n^2}\right)
\end{aligned}
\tag{27}
$$

By the fact that $\tilde{\mathcal{R}}_S(\bar{\mathbf{w}}_{\text{out}}) \leq 1$, we have:

$$
\mathbb{E}[\tilde{\mathcal{R}}_S(\bar{\mathbf{w}}_{\text{out}})] \leq \min_{\substack{\gamma \in \Gamma \\ S_{\text{out}} \in \mathcal{S}_{\text{out}}(\gamma)}} \tilde{\mathcal{O}}\left(\frac{1}{\gamma^2 n\varepsilon} + \frac{|S_{\text{out}}|}{\gamma n} + \frac{1}{n^2}\right) \wedge 1
\tag{28}
$$

Finally, apply Theorem 6.4,

$$
\mathbb{E}[\tilde{\mathcal{R}}_S(\bar{\mathbf{w}}_{\text{out}})] \leq \min_{\substack{S_{\text{out}} \subset S \\ \gamma(S \backslash S_{\text{out}}) > 0}} \tilde{\mathcal{O}}\left(\frac{1}{\gamma^2 n\varepsilon} + \frac{|S_{\text{out}}|}{\gamma n} + \frac{1}{n^2}\right) \wedge 1
\tag{29}
$$

$\square$

## L. Beyond unit norm assumption on data

For the ERM result, note that the dependence on $b$ arises solely through the sensitivity and projection dimension, as described in Lemma 6.3, Lemma D.1, and Lemma D.3. Combined with the adaptivity result established in Corollary G.2, we obtain:

**Theorem L.1** (Empirical risk bound for $\mathcal{M}^*$). *Running $\mathcal{M}^*$ with $\tilde{\mathcal{R}}_S$ satisfies $(\varepsilon, \delta)$-DP, and:*
*(1) For the averaged estimator $\bar{\mathbf{w}}_{\text{out}} \in \mathbb{R}^d$, we have utility guarantee in expectation:*

$$
\mathbb{E}_{\mathcal{A}}(\tilde{\mathcal{R}}_S(\bar{\mathbf{w}}_{\text{out}})) \leq \min_{\substack{S_{\text{out}} \subset S \\ \gamma(S \backslash S_{\text{out}}) > 0}} \tilde{\mathcal{O}}\left(\frac{b^2}{\gamma^2 n\varepsilon} + \frac{b|S_{\text{out}}|}{\gamma n} + \frac{1}{\varepsilon} + \frac{1}{n^2}\right) \wedge 1
$$

*(2) For the last-iterate estimator $\mathbf{w}_{\text{out}} \in \mathbb{R}^d$, w.p. $1 - 3/n^2$:*

$$
\tilde{\mathcal{R}}_S(\mathbf{w}_{\text{out}}) \leq \min_{\substack{S_{\text{out}} \subset S \\ \gamma(S \backslash S_{\text{out}}) > 0}} \tilde{\mathcal{O}}\left(\frac{b^2}{\gamma^2 n\varepsilon} + \frac{b|S_{\text{out}}|}{\gamma n} + \frac{1}{n\varepsilon} + \frac{1}{n}\right) \wedge 1
$$

For the population risk, the additional dependence on $b^2$ arising from the VC dimension can be absorbed into the ERM bound. Consequently, we have:

**Theorem L.2.** *Running $\mathcal{M}^*$ with $\tilde{\mathcal{R}}_S$ satisfies $(\varepsilon, \delta)$-DP, and w.p. $1 - 4/n^2$ over that last-iterate estimator $\mathbf{w}_{\text{out}} \in \mathbb{R}^d$:*

$$
\mathcal{R}_D(\mathbf{w}_{\text{out}}) \leq \min_{\substack{S_{\text{out}} \subset S \\ \gamma(S \backslash S_{\text{out}}) > 0}} \tilde{\mathcal{O}}\left(\frac{b^2}{n\gamma^2}(1 + \varepsilon^{-1}) + \frac{b|S_{\text{out}}|}{n\gamma} + \frac{1}{n\varepsilon} + \frac{1}{n}\right) \wedge 1
\tag{30}
$$

## M. Derivations for general distribution

### M.1. Settings

Without loss of generality, we assume that $\Theta$ contains a single item of interest, denoted as $\theta^*$. We upper bound the probability that $\theta^*$ is not selected, $\mathbb{P}(\theta^* \text{ is not selected}) = \text{PGF}(1 - 1/|\Theta|)$, by $\beta$, and then determine the corresponding distribution parameters. For notational convenience, let $t := 1/|\Theta|$.

## M.2. Upper bound on parameter $r$ for Truncated Negative Binomial distribution with $\eta > 0$

**Lemma M.1.** *Suppose $Q \sim \text{TNB}_{\eta,r}$ and $\mathbb{P}(\theta^* \text{ is not selected}) \leq \beta$, then $r \leq \frac{\beta^{1/\eta}}{|\Theta| + \beta^{1/\eta} - |\Theta|\beta^{1/\eta}}$*

We first do the relaxation:

$$\text{PGF}_{\text{TNB}(\eta,r)}(1-t) = \frac{(1-(1-r)(1-t))^{-\eta} - 1}{r^{-\eta} - 1}$$
$$\leq \frac{(r+t-rt)^{-\eta}}{r^{-\eta}}$$

Then we let:

$$\frac{(r+t-rt)^{-\eta}}{r^{-\eta}} \leq \beta$$
$$\Rightarrow r \leq \beta^{1/\eta}(r+t-rt)$$
$$\Rightarrow r \leq \frac{t\beta^{1/\eta}}{1 - \beta^{1/\eta} + t\beta^{1/\eta}} \qquad (31)$$
$$\Rightarrow r \leq \frac{\beta^{1/\eta}}{|\Theta| + \beta^{1/\eta} - |\Theta|\beta^{1/\eta}}$$

*Proof of Lemma 7.2.* For $\eta = 1$, we can do direct calculations as follows:

$$\text{PGF}_{\text{TNB}_{1,r}}(1-t) = \frac{\frac{1}{1-(1-r)(1-1/|\Theta|)} - 1}{1/r - 1}$$
$$= \frac{r(1-1/|\Theta|)}{1-(1-r)(1-1/|\Theta|)}$$
$$\leq \beta$$

which yields:

$$r(1-\beta)(1-1/|\Theta|) \leq \beta/|\Theta|$$
$$\Rightarrow r \leq \frac{1}{|\Theta|-1} \cdot \frac{\beta}{1-\beta}$$

$\square$

## M.3. Deferred discussion

In this section, we set the failure probability as $\beta = n^{-\alpha}$ for some $\alpha > 0$ and the hyperparameter set size as $|\Theta| = \log(n)$. From Lemma M.1, we obtain $\frac{1}{r} \geq \beta^{-1/\eta}|\Gamma| + (1-|\Gamma|) = (n^{\alpha/\eta} - 1)\log(n) + 1$, which implies that the expected number of repetitions should be at least $\tilde{\mathcal{O}}(n^{\alpha/\eta})$.

# N. Removal-Margin experiment

## N.1. Setting of the experiment

We train linear SVM classifiers on the CIFAR-10 dataset preprocessed by three methods: (1) pre-trained ViT features, (2) extracted features from pre-trained ViT, and (3) no preprocessing.

We iteratively eliminate "outliers," defined as data points with the smallest normalized margin value [5](Defn. N.1). After each removal, we refit the SVM and recompute the updated margin.

To account for variations in data scale across different classes, we plot the normalized margin, defined as follows:

**Definition N.1** (Normalized Margin). Given dataset $S$ and a linear classifier $h_{\mathbf{w}} : (\mathbf{x}, y) \mapsto \text{sign}(y\langle \mathbf{w}, \mathbf{x} \rangle)$, we define normalized margin w.r.t. $\mathbf{w}$ to be:

$$\text{Margin}(\mathbf{w}; S) = \max\left\{ \min_{(\mathbf{x},y) \in S} \frac{y\langle \mathbf{w}, \mathbf{x} \rangle}{\|\mathbf{x}\|\|\mathbf{w}\|}, 0 \right\} \qquad (32)$$

---

[5]before setting negative values to zero

## N.2. Results

While the margin remains at zero for the entire dataset, removing a few outliers reveals a clear trend: ViT-based features (green line) achieve a larger margin and grow more rapidly than ResNet-50-based features (red line) as more outliers are removed. Even after removing $1\%$ of the outliers, the dataset remains non-linearly separable, as indicated by the blue lines.

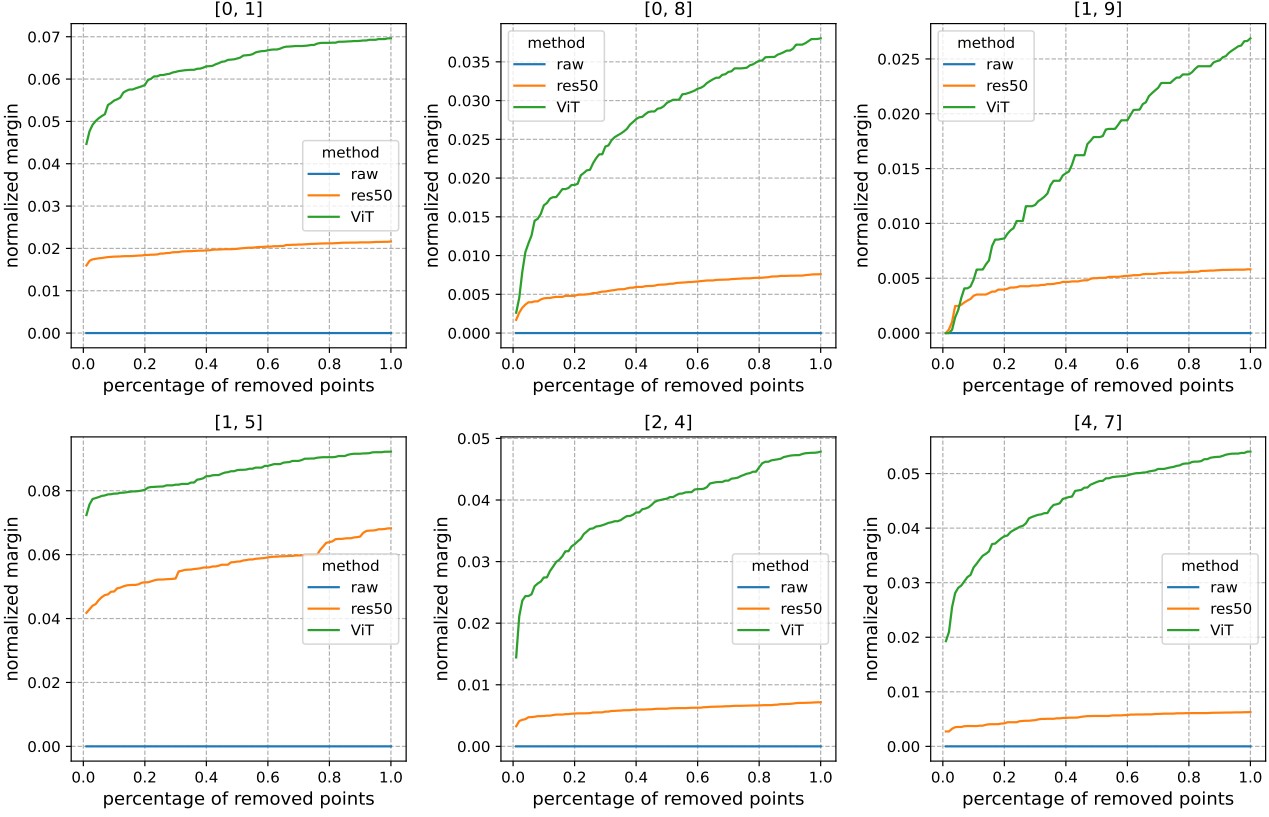

*Figure 3.* The number of removed points (in percentage of $n$) vs normalized margin. The classes from the CIFAR10 training set are labeled in each subtitle. As more points are removed, the margin increases.

