# OpenReview forum: "Adapting to Linear Separable Subsets with Large-Margin in Differentially Private Learning"
_ICML.cc/2025/Conference — ICML 2025 poster_

### Official Review · Reviewer_TPCg · 2025-03-08

**Overall Recommendation:** 4

**Summary:**

In this paper, the authors propose a $(\epsilon,\delta)$ differentially private algorithm for binary linear classification. The risk bound depends linearly on the arbitrary subset of data points $S_{out}$ , which if removed makes the data linearly separable with margin $\gamma$. The algorithm is adaptive as the knowledge of $\gamma$ or $S_{out}$ is not required by the algorithm.

 ### update after rebuttal
The authors have answered my questions and I would like to maintain my score.

**Claims And Evidence:**

Yes.

**Essential References Not Discussed:**

To the best of my knowledge essential references are mentioned.

**Experimental Designs Or Analyses:**

N/A

**Methods And Evaluation Criteria:**

N/A.

**Other Comments Or Suggestions:**

1. On line 572 (proof of Lemma A.4), $(1-t)^t$ should be $(1-r)^t$.

2. On line 602, $beta$ should be $\beta$.

3. In Algorithm 5, step 6, $\tilde{w}_t$ should be $w\_{t+1}$.

**Other Strengths And Weaknesses:**

The paper is very well written and makes important contributions.

1. When $|S_{out}|$ is small, the proposed work improves the risk bound of previous work by a factor of $\sqrt{n}$.

2. Knowledge of $\gamma$ or $S_{out}$ is not required.

3. Analysis of utility bound of advanced private hyperparameter tuning algorithm.

**Questions For Authors:**

Is the 3rd typo listed above indeed a typo? Otherwise, it is not clear how $w_t$ is being updated in Algorithm 5.

**Relation To Broader Scientific Literature:**

It improves the results over  Bassily et al. 2-22 and Nguyen et. al. 2020.

**Theoretical Claims:**

Yes. I checked supplementary material section A through F.

---

> ### Author Rebuttal · Authors · 2025-03-29
>
> Thank you for your detailed review and thoughtful question. We appreciate your recognition of our work. We have corrected all the typographical errors you identified.
>
> Regarding your question, you are indeed correct: $\tilde{w_t}$ in Algorithm 5 should indeed be $w_{t+1}$. We thank the reviewer once again for their careful reading and valuable feedback.

---

> > ### Comment · Reviewer_TPCg · 2025-04-02
> >
> > Thank you for your response. I will maintain my score.

---

> > > ### Author Response · Authors · 2025-04-04
> > >
> > > Thank you again for your careful review and feedback.

---

### Official Review · Reviewer_Ld9c · 2025-03-09

**Overall Recommendation:** 4

**Summary:**

The paper addresses the problem of DPERM for binary linear classification. The authors propose an efficient algorithm that achieves an empirical zero-one risk bound of $\widetilde{O}\left(\frac{1}{\gamma^2 \varepsilon n}+\frac{\left|S_{\text {out }}\right|}{\gamma n}\right)$. The algorithm is highly adaptive, requiring no prior knowledge of the margin parameter $\gamma$ or the outlier subset $S_{o u t}$. The paper also derives a utility bound for advanced private hyperparameter tuning. The main contributions include an efficient algorithm that adapts to largemargin subsets, an inlier-outlier analysis, and improved results in the agnostic case when the number of outliers is small.

## update after rebuttal
I am generally satisfied with the rebuttal and have raised my score.

**Claims And Evidence:**

The main claims are supported by proofs.

**Essential References Not Discussed:**

To the best of my knowledge, all necessary references are adequately discussed in the paper.

**Experimental Designs Or Analyses:**

Experiment details are less explained and the experiments mainly serves as a motivation of study.

**Methods And Evaluation Criteria:**

The proposed methodology is well-explained and aligns with intuitive expectations. However, I found the explanation of the experiments somewhat confusing. Please refer to my comments for further details.

**Other Comments Or Suggestions:**

- The experiments in the introduction appear confusing to me. I understand that the authors are trying to validate the idea that "a larger margin is believed to be the reason why pre-trained features help private learners to work better." However, the logic connecting better features, large margins, and the quick increase in performance when removing outliers seems somewhat incoherent. Additional explanation would be helpful.
- Please ensure the correct usage of \cite, \citep, and \citet. Note that the formatting of citations may vary depending on the LaTeX template used.
- Line 75, where is equation (2)? The position where the hyper-link jumps to does not have (2).
- Line 402, is $5m$ a typo?

**Other Strengths And Weaknesses:**

The writing of the paper is clear and fluent.

**Questions For Authors:**

- I am still a bit confused by Definition 6.1. As defined, $\mathcal{S_{in}} ( \gamma) $ is the collection of subsets that have $\gamma$ seperation. Can the author further explain why this exhaustive collection is necessary? Could something like $S_{out} = {\arg \min_{|S'|}} \gamma(S')\geq \gamma$ be defined instead? Also, as I understand, the choice for $S_{out}$ in the final bound can be arbitrary as long as it pertains $\gamma$ seperation. So I think some infimum of $|S_{out}|$ can be taken right? This is a minor point, but it seems to me that the bound depends on the selected subset, which makes it less conclusive.
- In Lemma 6.2, is the order of $k$ a worst case choice to guarantee that every $S_{in}$ in $\mathcal{S}_{in}(\gamma)$ has preserved margin?

**Relation To Broader Scientific Literature:**

The paper builds on prior work in differentially private half-space learning with large margins (e.g., Nguyên et al., 2020, and Bassily et al., 2022), with significant theoretical improvements. It also connects to the broader literature on neural collapse theory, which suggests that the last layer of a deep neural network trained on a classification task converges to distinct points.

**Theoretical Claims:**

The proofs are reviewed but not rigorously verified. Nevertheless, the results are consistent with intuition and appear to be sound.

---

> ### Author Rebuttal · Authors · 2025-03-30
>
> # Regarding your comments
> > "The experiments in the introduction....Additional explanation would be helpful."
>
> Thanks for asking this insightful question. Before presenting our explanations, we want to clarify that the “normalized margin”, labeled in the y-axis of Figure 2, measures the distance between decision boundaries. We believe you're essentially asking two related questions:
>
> **(1) Why do better feature representations imply a larger margin?**
>
> This analysis relates to neural collapse theory. Figure 2 in [WZSW 24] shows that improved feature representations yield smaller feature shift vectors—measuring deviation from ideal neural collapse features—indicating that empirical features align more closely with the equiangular tight frame structure and exhibit larger margins. Accordingly, ViT-pretrained features display smaller shift vectors than those from ResNet-50, reflecting a larger margin.
>
> **(2) How does it relate to the increase of margin as outliers got removed?**
>
> The intuition is that removing outliers makes classes linearly separable, yielding a positive margin. Continued removal of boundary-near points increases the normalized margin (y-axis, Figure 2) until it stabilizes near the data margin—a trend clearly seen in Figure 3 (top-left). Models with better feature representations show larger margins, explaining the ordering in Figure 2: ViT > ResNet-50 > Raw.
>
> [WZSW 24] Wang, C., Zhu, Y., Su, W. J., & Wang, Y. X. (2024). Neural collapse meets differential privacy: curious behaviors of NoisyGD with near-perfect representation learning. arXiv preprint arXiv:2405.08920.
>
> > Line 402, is 5m a typo?
>
> Thank you for bringing up this question. Our intended message is as follows: suppose we have $K$ hyperparameters, each with $m$ possible choices. Using $A_{\text{iter}}$ incurs an overhead that scales as $m^{\mathcal{O}(K)}$, whereas employing the advanced tuning methods leads to an overhead that scales as $\mathcal{O}(Km)$.
>
> > Other comments
>
> Thank you for pointing these out. We corrected the hyperlink and resolved the LaTeX \cite issues. Eq. 2 is defined as the geometric margin in Line 148.
>
> # Regarding your questions
> ## Question 1
> > “Why this exhaustive collection is necessary? Could something like … be defined instead?”
>
> Thank you for asking this thoughtful question. These definitions serve as tools for proving and stating the main theorem. For instance, they are directly used in the proofs of the margin preservation lemmas (Appendix B) and Lemma 6.4. (for proving adaptivity using a doubling grid).
> We found the definition you referenced is actually dependent on Definition 6.1. We assume you are referring to $\arg\min_{|S'|} \{ \gamma(S\/S') \geq \gamma \}$, which is equivalent to $ \arg\min_{S \in S_{\text{out}}(\gamma)} |S|$, where $S_{out}(\gamma)$ is defined in Definition 6.1.
>
> > “Also, as I understand, the choice for S_{out} … So I think some infimum can be taken right?”
>
> Yes, the infimum is indeed taken over all $S_{out}$ such that $\gamma (S \/ S_{out})>0$, as demonstrated in Theorems 6.5 and 6.7, specifically on the left-hand side of the inequality in each case.
>
> > “This is a minor point, but it seems to me that the bound depends..less conclusive”
>
> We appreciate your thoughtful question. Our bound is data-adaptive. We explain our result using Theorem 6.5 as an example.
>
> Since the dataset is finite, there must exist at least one optimal outlier subset $S_{out}^*$ that minimizes the upper bound:
> $\frac{1}{n\varepsilon\gamma(S_{out}^* )} + \frac{|S_{out}^*|}{n\gamma(S_{out}^*)}$
> While the minimizer of the upper bound depends on the optimal subset, this dependency reinforces the fact that the bound is data-adaptive. This stands in contrast to previous results, which are data-independent and yield a fixed rate of $n^{0.5}$, as reported in Table 1.
>
> Since this optimal subset is unknown—and identifying it via brute-force search is NP-hard—our algorithm, which runs in polytime can effectively adapt to it through a hyperparameter search over a logarithmic grid on the margin, as demonstrated in Theorem 6.5 and also pointed out by other reviewer.
>
>
> ## Question 2
> That's a good question. The order of k is determined by the JL lemma. The k is chosen to ensure every $\gamma$-level margin inlier set has preserved margin after projection with high probability.
>
> Specifically, from the statement of Lemma 6.2, the probability is taken over the randomness of the Johnson–Lindenstrauss matrix $\Phi$. It can be interpreted as the following conditional probability: $P_{\Phi} ( \gamma ( \Phi S_{in} ) \geq \gamma/3 \mid S_{in} ) \geq 1 - \beta$ rather than $P_{\Phi} (   \forall S_{in} \in \mathcal{S_{in}} (\gamma), \gamma ( \Phi S_{in}) \geq \gamma/3 ) \geq 1-\beta$. We note that the second inequality is a stronger condition, which is what you mentioned in the question. However, throughout this paper, the first one is sufficient for our proof.

---

### Official Review · Reviewer_1JSf · 2025-03-15

**Overall Recommendation:** 3

**Summary:**

This paper studies empirical risk minimization of large (geometric) margin half spaces, in the agnostic setting. They have the following major contributions:

a) They give an algorithm for this problem that works even without knowledge of the margin. Prior work by Nguyen et al. (2019) required knowledge of the margin $\gamma$. Their approach closely follows Nguyen's in this setting- they perform a JL transform to project the data down into lower dimension, and apply noisy SGD to learn in the lower dimension (arguing that the margin is preserved under the JL transformation, which significantly improves the dependence on the dimension). The main technical difference is that instead of requiring knowledge of the margin, they create a logarithmic discretized grid of the margin, and run the above base algorithm on all $\log n$ possible margin discretizations. They then noisily compare the average empirical risk in order to select the best margin (they also consider a version that uses private hyperparameter tuning developed in prior work by Talwar-Liu and Papernot-Steinke). They show that the bound of $O(1/\gamma^2 n \epsilon)$ with known margin can be matched even without knowing it.

b) To extend their result to the agnostic setting, they consider a notion of inliers (subsets of the datasets that are linearly separable with some margin), and outliers, and argue that a bound of $O(1/\gamma^2 \epsilon n + |S_{out}|/\gamma n)$ applies where $S_{out}$ is the set of outliers and $\gamma$ is the margin of the remaining points (note that this bound automatically applies for the best such subset $S_{out}$). They argue in their introduction that this is a practical way to think about margin, because for some learning problems (and some pretrained features), removing a few 'troublemaker' data-points results in much larger margin.

In addition to DP ERM, they also give similar bounds for the population risk version of the problem.

**Claims And Evidence:**

Yes, the proofs all seemed convincing and correct to me.

**Essential References Not Discussed:**

All essential references that I could think of were discussed.

**Experimental Designs Or Analyses:**

Not relevant.

**Methods And Evaluation Criteria:**

This is primarily a theoretical work, and so this is not a relevant section.

**Other Comments Or Suggestions:**

No other comments/suggestions

**Other Strengths And Weaknesses:**

Strength:
1. The definition of margin inliers and outliers, and developing a utility bound based on them seems like it could be useful, especially since the intro demonstrates that this phenomenon arises in natural learning settings. Additionally, unlike some prior work, this analysis technique also applies when the domain is bounded.

Weakness:
1. The main weakness of the paper is that the algorithm itself does not seem technically novel (the JL transform + GD approach has been used in prior work by Nguyen et al. and Bassily et al.), and the idea of choosing the margin by private selection over a grid was used in a different capacity (under a different margin definition) in Bassily et al. While the technical details vary slightly, I am not convinced that this algorithm is a significant contribution of this paper.
2. One issue is that for the bounds to be better than Bassily et al., the number of outliers needs to be relatively small (smaller than $O(\sqrt{n})$ where $n$ is the size of the dataset) whereas the intro discusses that in practical examples, a large portion (0.1% of the data) needs to be removed to achieve linear separability, so for large datasets, it's not clear that the utility bounds via this method would be better than existing bounds.

**Questions For Authors:**

I did not fully understand the relevance of the private hyperparameter tuning results- since the discretized hyperparameter set is sufficiently small (as the authors point out), is there really a need for this? Additionally, what parts of this analysis are novel- it seems like things should follow from the prior work on private hyperparameter tuning in a blackbox way.

**Relation To Broader Scientific Literature:**

This paper fits into the larger literature on DP learning theory- it considers the natural class of large margin half spaces studied in previous work (Nguyen et al. [2019], Bassily et al. [2022]) and gives utility bounds via a new notion of margin inliers and outliers. Prior work studied different notions of margin and/or assumed that the geometric margin was known.

**Theoretical Claims:**

I read through the important details of all of the proofs related to DP-ERM for this problem and am convinced that they are correct.

---

> ### Author Rebuttal · Authors · 2025-03-30
>
> # Regarding your comment on weaknesses
> > Weakness 1
>
> We thank the reviewer for their valuable feedback. We agree that our work builds upon the JL transform and gradient descent (GD) techniques, which have been explored in prior works such as Nguyen et al. and Bassily et al. However, our algorithm introduces several key innovations that go beyond a straightforward combination of these tools:
>
> **(1) Novel Margin Definition and Analysis:**
> Our analysis is based on a new definition of margin inliers/outliers, which, to the best of our knowledge, has not appeared in prior work. This definition is data-dependent (as discussed in Remark 3.1), in contrast to the confidence margin used in Bassily et al., which is data-independent. This distinction is not only conceptual but also technical: our data-dependent margin enables a new analysis technique that is not limited to the JL + GD setting (please refer to comments from Line 291 to Line 295). Most importantly, it leads to a data-adaptive generalization bound that avoids the hard $1/\sqrt{n}$ rate in the agnostic case that appears in Bassily et al.
>
> **(2) Practical and Efficient Private Margin Selection:**
>
> While Bassily et al. employs a grid search combined with the exponential mechanism, their approach involves a more complex score function. Additionally, the exponential mechanism appears hard to implement, as evaluating the score function requires solving non-convex optimization problems (Lemma F.3). In comparison, our method is computationally efficient and straightforward to implement.
>
>
> > Weakness 2
>
> Thank you for the insightful question. To address it more generally, we note that the problem of tolerating a constant fraction of outliers—under certain additional assumptions on the noise model—remains open, as discussed in Section 8. We have also included a comparison between our bound and that of Bassily et al. in Lines 165 to 168.
>
>
>
> # Regarding your question
>
> Thank you for this thoughtful question. First, we clarify that this section extends our main result by analyzing utility in the context of general hyperparameter sets, which may not be small. For completeness, we provide a utility analysis of hyperparameter tuning. Specifically, advanced tuning methods yield a utility bound with a $\log(|\Theta|)$ dependence, whereas naively evaluating all base mechanisms and selecting the best incurs a $|\Theta|$ dependence. However, in the small hyperparameter setting considered in our paper, the utility bound is not dominated by the $\log\log(n)$ as it is obscured by other $\log(n)$ factors (see Section 7.2). In addition, to the best of our knowledge, the explicit form of the utility bound has not been previously derived.
>
>
>
> Lê Nguyễn, Huy, Jonathan Ullman, and Lydia Zakynthinou. "Efficient private algorithms for learning large-margin halfspaces." Algorithmic Learning Theory. PMLR, 2020.
>
> Bassily, Raef, Mehryar Mohri, and Ananda Theertha Suresh. "Differentially private learning with margin guarantees." Advances in Neural Information Processing Systems 35 (2022): 32127-32141.

---

### Decision · Program_Chairs · 2025-05-01

**Decision:**

Accept (poster)

**Comment:**

This paper addresses empirical risk minimization for large-margin half-spaces under differential privacy, proposing a margin-adaptive method that does not require prior margin knowledge. The reviewers appreciate the introduction of a novel definition distinguishing margin inliers and outliers, offering improved bounds especially beneficial when the outlier fraction is small. However, concerns were raised about the technical novelty, as the core algorithmic techniques (Johnson-Lindenstrauss transform combined with noisy gradient descent and margin selection via discretization) closely follow prior work. Authors effectively clarified the unique contributions in their rebuttal, emphasizing the data-dependent margin definition and practical efficiency improvements over existing methods. Given the support for the paper, we recommend acceptance.